# Foundation Cures Personalization: Improving Personalized Models' Prompt Consistency via Hidden Foundation Knowledge

**Yiyang Cai**[1], **Zhengkai Jiang**[2], **Yulong Liu**[1], **Chunyang Jiang**[1]
**Wei Xue**[1], **Yike Guo**[1], **Wenhan Luo**[1*]
[1] Hong Kong University of Science and Technology (HKUST)
[2] Tencent Hunyuan
`https://yiyangcai.github.io/freecure-aigc.github.io/`

## Abstract

Facial personalization faces challenges to maintain identity fidelity without disrupting the foundation model's prompt consistency. The mainstream personalization models employ identity embedding to integrate identity information within the attention mechanisms. However, our preliminary findings reveal that identity embeddings compromise the effectiveness of other tokens in the prompt, thereby limiting high prompt consistency and attribute-level controllability. Moreover, by deactivating identity embedding, personalization models still demonstrate the underlying foundation models' ability to control facial attributes precisely. It suggests that such foundation models' knowledge can be leveraged to **cure** the ill-aligned prompt consistency of personalization models. Building upon these insights, we propose **FreeCure**, a framework that improves the prompt consistency of personalization models with their latent foundation models' knowledge. First, by setting a dual inference paradigm with/without identity embedding, we identify attributes (*e.g.*, hair, accessories, etc.) for enhancements. Second, we introduce a novel foundation-aware self-attention module, coupled with an inversion-based process to bring well-aligned attribute information to the personalization process. Our approach is **training-free**, and can effectively enhance a wide array of facial attributes; and it can be seamlessly integrated into existing popular personalization models based on both Stable Diffusion and FLUX. FreeCure has consistently shown significant improvements in prompt consistency across these facial personalization models while maintaining the integrity of their original identity fidelity.

## 1 Introduction

Human face-centric personalization represents compelling downstream applications of art creation, advertising, and entertainment in the realm of text-to-image synthesis [23, 50]. Given a limited number of images that depict particular identities, facial personalization models generate novel content that reflects these identities through diverse conditions [14, 46, 69, 38]. However, this aspiration is hindered by a persistent challenge: the necessity to maintain high fidelity to the identity while ensuring the controllability of the generated content, also referred to as prompt consistency [72, 26]. This challenge is pronounced in facial personalization since any imperfections or misalignment in the generated faces are particularly salient to humans. Therefore, compared to common object personalization, human face-centric personalization mandates dedicated attention and research efforts.

---

*Corresponding author

39th Conference on Neural Information Processing Systems (NeurIPS 2025).

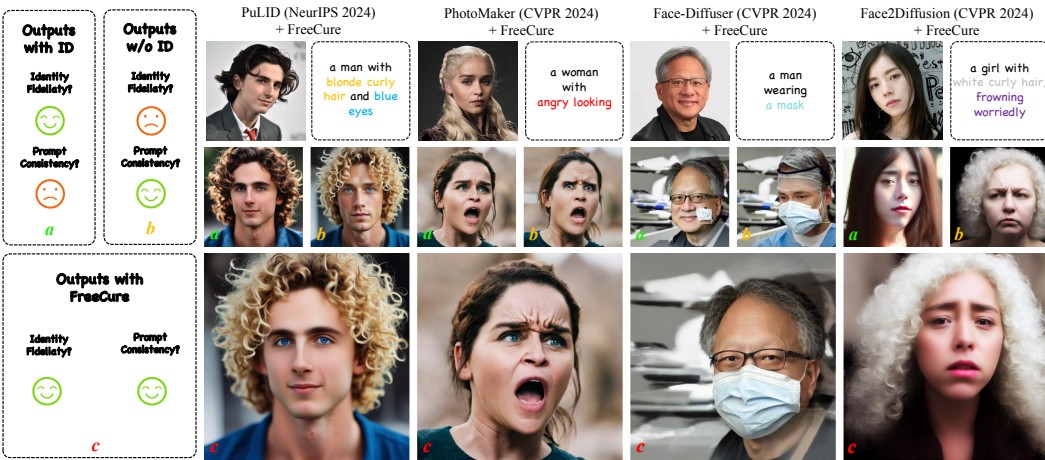

Figure 1: Personalization models (a) demonstrate strong capability in preserving identity fidelity, albeit at the cost of reduced prompt consistency. A prevalent feature in most personalization models is that when their identity embedding inputs are deactivated, they regain the ability to exhibit highly accurate prompt consistency with respect to facial attributes (b), a characteristic closely aligned to their foundation models. Our proposed *FreeCure* effectively leverages the latent foundational knowledge inherent in personalized models, enhancing prompt consistency in scenarios involving complex facial attribute control while preserving the identity fidelity (c).

Previous research of facial personalization aims to integrate identity-specific information into the cross-attention mechanisms, with either fine-tuning based strategy [46, 14] or tuning-free paradigm with an identity encoder [64, 34, 18, 59, 27]. However, the dual objectives of maintaining prompt consistency and identity fidelity remain unresolved. Despite the aforementioned advancements, state-of-the-art personalization techniques still struggle to enhance identity fidelity without sacrificing prompt consistency.

Our preliminary experiments indicate that, under identical experimental conditions (*e.g.*, prompts and initial random noise), personalization models' outputs exhibit a significant decline in prompt consistency compared to generated results without identity embedding. For instance, as shown in the first two examples in Fig.1, personalization models fail to generate *"blonde curly hair"* and *"angry"* faces accurately, whereas their counterparts without identity embedding can handle these prompts in a highly faithful manner. Plus, when handling more complex prompts that consist of multiple attributes, the personalization model generates even poorer results. These findings show a fact overlooked by most previous works: while personalization models show their degradation in prompt consistency, the ability of their original foundation models **is unharmed but overridden**. Based on this observation, we further find that identity embeddings can significantly undermine prompt consistency by impeding normal representation of other attribute-related tokens within the prompt through cross-attention mechanisms. This, in turn, adversely affects their effective expression in the latent space of the U-Net. However, given the compelling zero-shot identity extraction capability of identity embedding, directly modifying personalization models' well-trained cross-attention modules can destroy their ability to capture precise identity information. Therefore, our core pursuit is clear: *Is it feasible to mitigate the erosion of prompt consistency in personalization models while keeping their trained cross-attention modules unaffected?*

Motivated by this objective, we propose FreeCure, a framework enhancing the prompt consistency of personalization models through the guidance of their latent foundation knowledge. While keeping cross-attention modules intact, we propose a novel foundation-aware self-attention (FASA), enabling attributes with high prompt consistency to replace those that are ill-aligned during personalization generation. To protect the identity unharmed, this strategy also leverages semantic segmentation models to generate the scaling masks of these attributes, therefore making such replacement happen in a highly localized and harmonious manner. Furthermore, we use a simple but effective approach called asymmetric prompt guidance (APG) to restore abstract attributes such as expression. Through comprehensive experiments, FreeCure has been verified to effectively restore a wide variety of misaligned attributes produced by various state-of-the-art personalization models.

In summary, the contributions of our paper are threefold:

- We identify the limitations in prompt consistency prevalent across existing face-centric personalization models. Building upon this, we explore the negative effects of identity embedding and elucidate the fundamental reasons for its adverse impacts.
- We propose a training-free framework that leverages high prompt consistency information from foundation models to enhance multiple weakened attributes generated by personalization models. The enhancement of various attributes is achieved without mutual interference.
- Our framework can be seamlessly integrated into widely used personalization models and diverse foundational models, including Stable Diffusion and FLUX. Comprehensive experiments show that our approach improves prompt consistency while maintaining the well-trained ability for identity preservation.

## 2 Related Work

**Identity-preserving Generation.** Identity-preserving generation can be broadly categorized into fine-tuning-based and encoder-based approaches. Fine-tuning-based methods [14, 57, 3, 19, 73, 12, 67, 40] either optimize a vector within a textual embedding to encode identity, modify specific weights in the model [46, 31, 7, 25, 4, 17, 47, 52] or use LoRA techniques [24]. However, these methods require training a distinct model for each identity, which compromises scalability and increases susceptibility to overfitting. In contrast, encoder-based methods utilize large-scale pretraining to automatically derive identity representations aligning with textual embeddings, enabling zero-shot personalization with novel identity references. Some methods [63, 9, 64, 49, 62, 34, 43] only train a mapping network or modify cross-attention weights to embed identity information, while others [65, 15, 60, 36, 59, 56, 33, 71, 10] incorporate cross-attention adapters during training. Recent works [18, 27] also leverages Diffusion Transformers (DiT) [1, 42, 13] to reach more impressive performance. In this work, we focus primarily on encoder-based methods, as they represent the state-of-the-art paradigm for personalization and demonstrate particular efficacy in facial performance.

**Attention in Diffusion Models.** Attention mechanisms serve as foundational components in text-to-image diffusion models. Prior research [20, 16, 74, 30, 55, 6, 35, 29, 37, 41] has leveraged semantic information from cross-attention maps to facilitate object-level editing, while other methods [54, 2, 39, 8, 61, 21, 11, 53, 51, 58] have employed spatial features from self-attention layers to achieve more precise modifications or style transfer. However, the positional information derived from attention maps are inherently constrained to object-level manipulation, making them less robust for fine-grained facial attribute generation. Plus, the function of face-centric embeddings within attention mechanisms remains insufficiently explored, especially in facial personalization models.

## 3 Revisit ID Embedding in Personalization

We conduct a comprehensive analysis to justify the limitations of current personalization methods in maintaining prompt consistency. This investigation highlights the challenges inherent in existing approaches provides a foundational basis and critical insights for our proposed methodology.

### 3.1 Dual Denoising to Study Prompt Consistency

To elucidate current personalization methods' limitations, we devise a comparative experiment as an initial exploration. We maintain a fixed noisy latent code $z_T$ and implement two parallel denoising procedures [5]. The only difference is that one denoising procedure incorporates textual embeddings $c$ and identity embedding $c_{id}$, while the other set $c_{id}$ into a zero tensor $\tilde{c_{id}}$. For clarity, we will refer to the denoising procedure with identity embedding (personalization denoising) as **PD**, and the denoising process that excludes identity embedding (foundation denoising) as **FD** hereafter:

$$\textbf{PD} : \epsilon_p = \epsilon_\theta(z_t, t, c, c_{id}); \textbf{FD} : \epsilon_f = \epsilon_\theta(z_t, t, c, \tilde{c_{id}}) \tag{1}$$

where $\epsilon_p$ and $\epsilon_f$ denote the predicted noise from PD and FD, respectively. We have conducted these experiments using two facial personalization methods [64, 34] and results are presented in Column

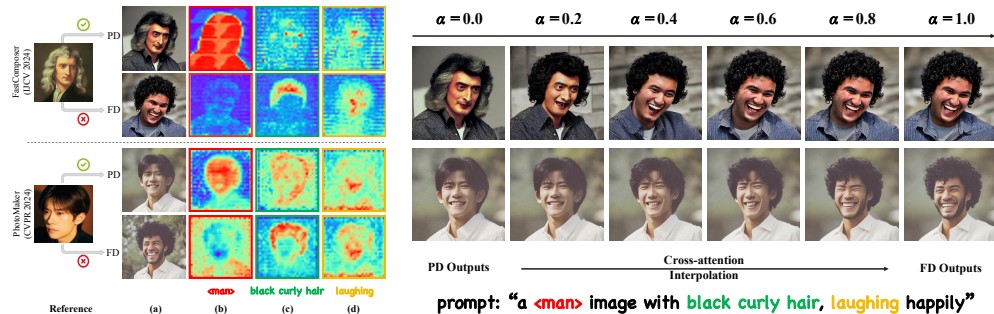

Figure 2: **Analysis on cross-attention maps of facial personalization models. Left**: token-wise attention map visualization. **Right**: interpolation experiment on PD and FD's cross-attention maps.

(a) of Fig.2. It is evident that the output from PD exhibits a reduced ability to match the specific facial attributes compared to the counterpart without identity embedding. For instance: 1) The hair features do not align with the prompt's specified "*black curly*" as expected; 2) The expression "*laughing*" is also restricted. Conversely, the results from FD show greater prompt consistency. These findings indicate that current identity embedding methods may not effectively address "the balance between identity fidelity and prompt consistency".

### 3.2 ID Embedding's Effect on Attention Layers

**OBSERVATION 1: ID embeddings disrupt cross-attention layers.** To investigate the underlying causes of this phenomenon, we conduct a visualization experiment inspired by [20, 2, 54], because identity embedding is primarily fused into cross-attention layers. Columns (b-d) of Fig.2's left part present the visualization results of cross-attention maps for both PD and FD processes, focusing on identity embeddings and the attribute-specific token embeddings for "*black curly hair*" and "*laughing*". It reveals that the identity embedding significantly amplifies activation in facial regions while simultaneously reducing activation for other tokens and disrupting their typical interactions within the cross-attention mechanism. Additionally, we observe that during the FD process, prompt consistency remains preserved. This observation underscores an insight: although most personalization models are well fine-tuned, their intrinsic capability to generate faces with high prompt consistency is still preserved but unexpectedly overridden by external identity embeddings. Notably, this latent capability can be effectively activated via the FD procedure, as its name "Foundation Denoising" indicates.

**OBSERVATION 2: Personalized cross-attention layers are highly susceptible.** We conduct another experiment on cross-attention layers, which involves incorporating a portion of the cross-attention maps from the FD process into those of the PD process. The formula for this approach is

$$A^p_{:,:,m} \leftarrow Softmax(\alpha A^f_{:,:,n} + (1 - \alpha)A^p_{:,:,m}), \tag{2}$$

where $A^p_{:,:,m}$ represents the cross-attention map of PD's identity embedding, and $A^f_{:,:,n}$ represents the cross-attention map of the FD's zero-valued embedding. The parameter $\alpha$ controls the weight of the FD's cross-attention map that is injected. The results are shown on the right part of Fig.2. It is observed that, as the weight of the map from the FD process increases, the model quickly loses identity fidelity, even though it regains the facial attributes that were missing before. This experiment demonstrates that cross-attention layers are rather susceptible. To preserve the models' capacity for identity preservation, it is better to leave these well-trained cross-attention modules unaltered.

Our preliminary findings demonstrate that identity embeddings in cross-attention layers are disrupting personalization models' prompt consistency. In contrast, its strong ability of identity extraction makes it rather challenging to be modified. Given this "dilemma", we aim to explore a new approach from the perspective of self-attention, inspired by [16, 52]. Since most personalization models introduce minimal modifications to self-attention layers, it is reasonable to assume that the aforementioned hidden foundation knowledge is preserved within them. By enhancing the self-attention layers in personalization models while keeping cross-attention layers intact, we anticipate achieving better alignment of facial attributes in personalized generation.

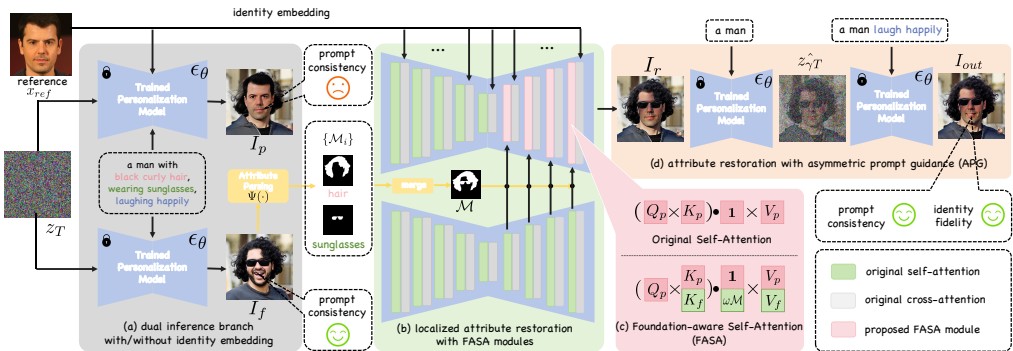

Figure 3: **Overview of FreeCure.** For a personalization model $\epsilon_\theta$, we first introduce **(a)**: dual inference paradigm to generate faces with/without identity ($I_p$ and $I_f$), where $I_f$ without identity embedding shows better prompt consistency. Next, we leverage a segmentation model $\Psi(\cdot)$ to derive related masks of target attributes with clear spatial information (hair, sunglasses, etc.) and merge them into a mask $\mathcal{M}$. In **(b)**: we modify the original self-attention modules with our proposed FASA **(c)**, which concatenates key and value matrices of FD process and PD process, together with a scaling mask to achieve the attribute injection. Finally, we utilize a simple yet effective strategy **(d)**: asymmetric prompt guidance (APG) to restore abstract attributes (*e.g.*, expressions).

## 4 Methodology

Building upon our investigation in Sec.3, we propose FreeCure, a training-free framework designed to improve the prompt consistency of facial personalization models (see Fig.3). Initially, we develop a foundation-aware self-attention (FASA) mechanism to integrate localized attributes from the FD process into the PD process (Sec.4.1). Subsequently, we apply an asymmetric prompt guidance (APG) strategy to reconstruct more abstract attributes (Sec.4.2).

### 4.1 Foundation-Aware Self-Attention

Given a reference image $x_{ref}$ that provides the target identity, a well-trained personalization model $\epsilon_\theta$, a user-defined prompt $c$ that contains a sequence of facial attributes $\mathcal{A} = \{A_1, A_2, \cdots A_n\}$, our goal is to employ knowledge in FD to enhance prompt consistency of PD's output.

**Obtain Masks of Spatial Localized Attributes.**
As shown in Fig.3 (a), we adapt the dual inference branches which are identical to the FD/PD process introduced in Sec.3 for personalization models to generate the personalized face $I_p$ with unsatisfactory prompt consistency as well as the foundation face $I_f$ with high prompt consistency. Next, we utilize an external face parsing model, denoted as $\Psi(\cdot)$, to extract the binary mask $M_i$ corresponding to $A_i$ from the foundation outputs $I_f$. Generally, when restoring multiple attributes, we simply need to compute the different masks respectively and merge them $\mathcal{M} = \bigcup\{M_i\}$. $\mathcal{M}$ contains the spatial information of attributes that align with the target prompt, which will play an important role in the attribute restoration process that will be mentioned in the next part.

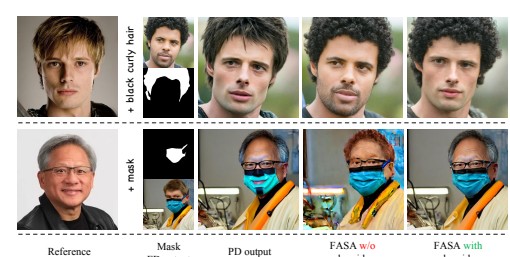

Figure 4: **Fine-grained attribute enhancement via masks.** Extracting masks from the FD results makes the FASA module only focus on enhancement for target attributes, minimizing its negative effect on identity fidelity.

**FASA Mechanism.** FASA is the core module that links the information between FD and PD processes and enables personalization models to restore the attributes via their foundation knowledge. As shown in Fig.3(c), we at first identify the self-attention modules in the model. Specifically, in each timestep $t$ and attention layer $l$, we denote PD and FD's key, query, and value matrices as $\mathcal{KQV}_p^{tl} = \{K_p^{tl}, Q_p^{tl}, V_p^{tl}\}$ and $\mathcal{KQV}_f^{tl} = \{K_f^{tl}, Q_f^{tl}, V_f^{tl}\}$. Second, we concatenate the key and

value matrices of FD process to those of PD process: $\hat{K}^{tl} = [K_p^{tl}, K_f^{tl}], \hat{V}^{tl} = [V_p^{tl}, V_f^{tl}]$. Thus, by omitting the labels of $t$ and $l$ for simplicity, the operation of FASA is

$$\text{FASA}(\mathcal{KQV}_p, \mathcal{KQV}_f) = \text{Softmax}(\frac{Q_p\hat{K}^T}{\sqrt{d}})\hat{V}. \tag{3}$$

**Fine-grained Restoration with Scaling Masks.** The approach described in Eq.3 allows the PD process to obtain information from the FD process. However, we observe that the method also retains a substantial degree of unrelated features, leading to a notable loss of identity fidelity, as shown in Fig.4. To constrain attribute restoration in a specific region without disrupting identity information, we apply pre-calculated masks $\mathcal{M}$ of different attributes to the similarity map. Additionally, to further enhance FASA's performance, we introduce an additional scaling factor, denoted as $\omega$, to control the magnitude of injecting attribute features from FD to PD. Therefore, the enhanced FASA mechanism can be written as

$$\text{FASA}(\mathcal{KQV}_p, \mathcal{KQV}_f) = \text{Softmax}(\frac{[\mathbf{1}, \omega\mathcal{M}] \odot Q_p\hat{K}^T}{\sqrt{d}})\hat{V}. \tag{4}$$

Where $\mathbf{1}$ denotes a matrix with all elements equal to 1, designed to preserve attention to the original features within the PD process. $\odot$ represents the Hadamard product.

In the full-attention layers of Diffusion Transformers (DiT) such as FLUX, visual and textual information is integrated into a unified sequence representation $[X; C]$. Within this architecture, the FASA mechanism operates in a similar manner, with a key distinction: the attribute mask is applied solely to the component derived from the visual queries of the PD branch ($Q_p^X$) and the visual keys of the FD branch ($K_f^X$). This selective masking strategy preserves the original cross-modal attention patterns between visual and textual elements, resembling OminiControl [51]:

$$\text{FASA}_{flux}(\mathcal{KQV}_p, \mathcal{KQV}_f) = \text{Softmax}(\frac{\mathcal{M}(\omega)_{flux} \odot Q_p\hat{K}^T}{\sqrt{d}})\hat{V}, \tag{5}$$

$$\mathcal{M}(\omega)_{flux} = \begin{pmatrix} \mathbf{1}_{l_1 \times l_1} & \mathbf{1}_{l_1 \times l_2} & \omega\mathcal{M}_{l_1 \times l_1} \\ \mathbf{1}_{l_2 \times l_1} & \mathbf{1}_{l_2 \times l_2} & \mathbf{0}_{l_2 \times l_1} \end{pmatrix} \tag{6}$$

Where $Q_p = [Q_p^X; Q_p^C]$ is the original query matrix of PD branch, and $\hat{K} = [K_p^X; K_p^C; K_f^X], \hat{V} = [V_p^X; V_p^C; V_f^X]$ are FASA's enhanced key and value matrices which integrates information from both PD and FD branches. With such enhancement, attributes of the FD process can be successfully extracted and precisely injected into the PD process without disrupting the personalization model's ability of maintaining identity fidelity.

## 4.2 Asymmetric Prompt Guidance

Following attributes with clear location restored via FASA, to enhance more abstract attributes such as expressions, we introduce a simple yet effective method called *asymmetric prompt guidance* (APG). This strategy is based on the diffusion model's inversion [50]. During the inversion phase, the model accepts only a template prompt (e.g. *"a man"*) that does not include any target attributes. In the denoising process, we add the target attributes' tokens back to this template prompt (e.g. *"a man"* $\rightarrow$ *"a man laughing"*). By leveraging the pretrained controllability of the foundation model, this approach enhances such attributes, resulting in the final refined image $I_{out}$. Throughout the denoising process, we use only pure textual prompts without identity embeddings, thereby avoiding their potential influence on the tokens related to the target attributes, as discussed in Sec.3. Furthermore, to better preserve the identity, we start the denoising process directly from an intermediate latent code $z_{\hat{\gamma}T}, \gamma \in [0, 1]$, where the high-level identity information has already been established.

## 5 Experiments

**Evaluation Datasets.** We collect an extensive dataset including 50 identities, with 30 derived from the CelebA-HQ [32] and the other 20 non-celebrity identities curated by our team. Each identity is represented by a single image, with a spectrum of facial characteristics. The prompt set consists of 20 prompts containing different facial attributes. For each (*identity, prompt*) pair, we produce 20 images. Detailed information can be found in Appendix.A.2.

**Baselines.** We evaluate FreeCure using several representative facial personalization methods: FastComposer [64], Face-diffuser [62], Face2Diffusion [49], InstantID [59], PhotoMaker [34], PuLID [18], and InfiniteYou [27]. Among these, Face-diffuser, Face2Diffusion, and FastComposer are implemented using SDv1-5 [45], whereas InstantID, PhotoMaker, and PuLID are based on SD-XL [44]. PuLID and InfiniteYou employ FLUX.1-dev [1] as their foundation models.

**Evaluation Metrics.** We adopt CLIP-T [22] to calculate prompt consistency (**PC**). To calculate identity fidelity (**IF**), we use MTCNN [68] and FaceNet [48] to extract the embedding of the generated/reference faces and compute the cosine similarity. Following PhotoMaker, we adopt the face diversity (**Face Div.**) metric which calculates LPIPS [70] between facial areas. Lastly, following Face2Diffusion, we adopt **PC** $\times$ **IF** score, since it reflects the overall balance of prompt consistency and identity fidelity. We compute the harmonic mean (hMean) of PC and IF.

**FreeCure Settings.** We set $\omega$ in FASA to 2.0, to ensure that attribute information from FD can be sufficiently integrated into PD. The $\gamma$ in APG is set to 0.5 to maintain the balance between identity fidelity and prompt consistency. For attribute segmentation, we leverage BiSeNet [66] and Segment-Anything [28] for different facial attributes.

## 5.1 Main Results

Table 1: **Main quantitative evaluation results.** With FreeCure, the mainstream personalization models' prompt consistency is highly enhanced on critical quantitative metrics.

| Method | PC(%) ↑ | IF(%) ↑ | Face Div. (%) ↑ | PC × IF (hMean) ↑ |
|---|---|---|---|---|
| FastComposer | 18.14 | **43.19** | 38.92 | 25.55 |
| FastComposer + **FreeCure** | **21.02** (+15.91%) | 41.02 (-5.02%) | **41.01**(+5.37%) | **27.80** (+8.82%) |
| Face-Diffuser | 20.67 | **58.34** | 40.82 | 30.52 |
| Face-Diffuser + **FreeCure** | **22.48** (+8.76%) | 57.51 (-1.42%) | **41.95**(+2.77%) | **32.32** (+5.90%) |
| Face2Diffusion | 21.92 | **39.98** | 43.51 | 28.31 |
| Face2Diffusion + **FreeCure** | **23.26** (+6.12%) | 39.23 (-1.88%) | **44.29**(+1.79%) | **29.20** (+3.15%) |
| InstantID | 21.89 | **63.94** | 48.98 | 32.61 |
| InstantID + **FreeCure** | **23.62** (+7.90%) | 62.01(-3.02%) | **51.82** (+5.80%) | **34.21** (+4.91%) |
| PhotoMaker | 23.04 | **51.84** | 47.29 | 31.90 |
| PhotoMaker + **FreeCure** | **24.91** (+8.11%) | 50.15 (-3.26%) | **48.52** (+2.60%) | **33.28** (+4.34%) |
| PuLID (SDXL) | 25.16 | **58.23** | 42.12 | 35.14 |
| PuLID (SDXL) + **FreeCure** | **26.05** (+3.55%) | 56.95 (-2.20%) | **43.72** (+3.80%) | **35.74** (+1.74%) |
| PuLID (FLUX) | 22.42 | **74.97** | 43.91 | 34.52 |
| PuLID (FLUX) + **FreeCure** | **24.78** (+10.53%) | 72.61 (-3.15%) | **46.09** (+4.96%) | **36.95** (+7.04%) |
| InfiniteYou | 23.77 | **79.71** | 44.28 | 36.62 |
| InfiniteYou + **FreeCure** | **25.25** (+6.23%) | 77.13 (-3.24%) | **46.82** (+5.74%) | **38.05** (+3.90%) |

**Overall Performance Comparison.** Table 1 and Fig. 5 & 6 present a comparative analysis of our proposed method with the baselines, examining both quantitative and qualitative aspects. It is easy to notice that baselines often fail to accurately reflect key facial attributes mentioned in the prompts. For instance, these baselines often generate faces with identical expression (row 2, 4 of Fig.5 and row 2 of Fig.6). For attributes with large areas (e.g., hair, sunglasses), baselines often cannot generate them harmoniously (row 3, 4 and 5 of Fig.5 and row 1 of Fig.6). Conversely, our approach shows a remarkable ability to enhance absent or faint attributes, significantly improving prompt consistency for these baselines. FreeCure can even tackle subtle facial attributes (e.g., eye color and earrings, see row 5, 6 of Fig.5). Notably, FreeCure achieves significant performance improvements across all foundation models' personalization methods, demonstrating its strong generalizability. Secondly, we notice that FreeCure leads to a slight decline in identity fidelity, which can be attributed to positively improved facial diversity. Since baselines tend to produce faces closely resembling their references, they inherently score higher in identity fidelity. Ultimately, in terms of the PC $\times$ IF metric, our method also shows considerable improvement over all baselines. In conclusion, both quantitatively and qualitatively, FreeCure demonstrates a positive balance by enhancing prompt consistency while keeping the reduction in identity fidelity to a minimum. Additional comparisons and results on more references (including non-celebrities) are available in Appendix. A.3.1. We also

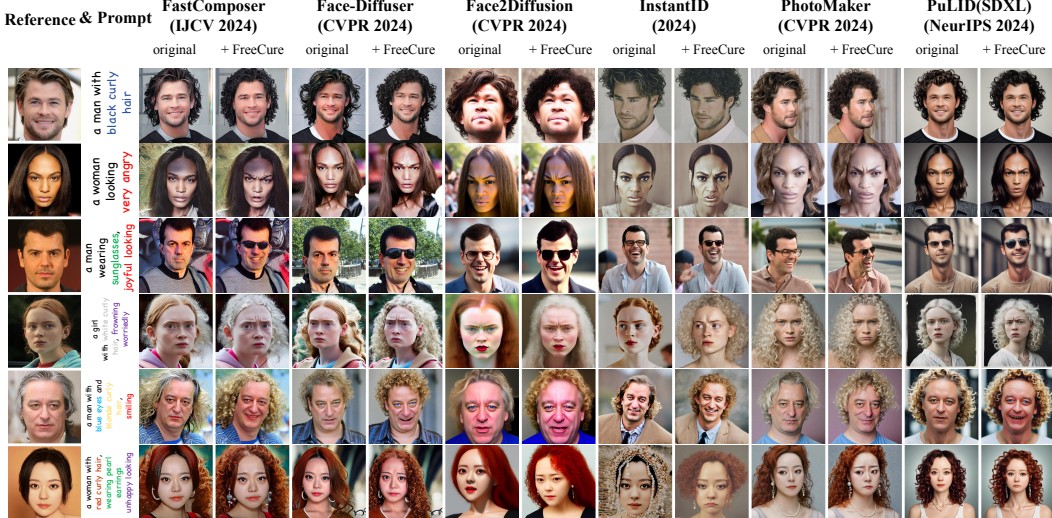

Figure 5: **Qualitative comparison with facial personalization baselines (including baselines built upon SDv1.5 and SDXL).** Different attributes in prompts are highlighted in various colors. Comparison of corresponding FD outputs is provided in the Appendix.A.3.2.

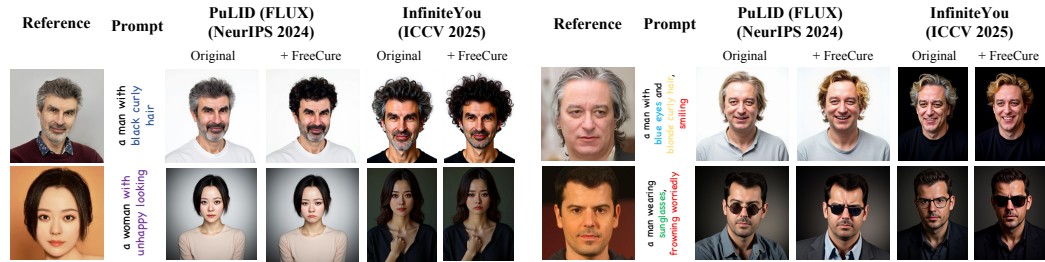

Figure 6: **Qualitative comparison with FLUX-based facial personalization baselines.** Different attributes in prompts are highlighted in various colors.

provide corresponding FD outputs for Fig.5 to show the effectiveness of FreeCure to transfer correct attributes from the FD to PD process in Appendix. A.3.2

**Prompt Consistency with Multiple Attributes.** Prompts involving multiple facial attributes pose a greater challenge to personalization's prompt consistency but better reflect practical user needs. Table 2 illustrates FreeCure's improvements on baselines with prompts including multiple facial attributes. As the number of attributes increases, the PC values of baselines tend to decrease. In contrast, FreeCure's improvement in PC becomes more significant as the complexity of the prompt increases. More analysis of FreeCure's non-disruption manner in multiple prompt personalization is available in Appendix.A.4.1. In summary, through integration with FreeCure, personalization models can effectively address more complex and realistic prompt instructions.

## 5.2 Robustness Justification

**Robust Performance with Different Initial Noises.** We observe that, with identical initial noise, all baselines' FD and PD processes generate faces with similar attribute locations. It is important to validate that FASA's robust performance without these condition. Thus, we relax this condition and regenerate faces with different initial noises. Fig.7 shows that results under the two settings are comparable, which confirms that FASA can effectively enhance the generated results of PD, even when its FD counterpart produces faces with variable spatial structures.

**Visualization of FASA maps.** Fig.8 visualizes the FASA map $A \in \mathbb{R}^{H \times (2 \times W)}$, whose size is doubled according to Eq.4. For a given query point $q_i$ in $Q_p$, its corresponding scores $A_{i,:}$ are extracted from both $K_p$ and $K_f$. When $q_i$ falls into areas associated with target attributes (*e.g.*,

Table 2: **Quantitative comparison of prompt consistency with different number of attributes.** After calculating the metrics for each method, we compute their mean values based on the corresponding foundation model type. For metrics of each baseline, please refer to Appendix.A.3.3.

| Foundation Model | PC (1 Attr.) ↑ | PC (2 Attr.) ↑ | PC (3 Attr.) ↑ |
|---|---|---|---|
| SDv1.5 | 21.01 | 20.34 | 18.49 |
| SDv1.5 + **FreeCure** | **22.70** (+8.04%) | **22.34** (+9.87%) | **21.16** (+14.44%) |
| SDXL | 23.83 | 23.31 | 22.65 |
| SDXL + **FreeCure** | **25.11** (+5.34%) | **24.80** (+6.36%) | **24.49** (+8.14%) |
| FLUX | 24.15 | 22.64 | 21.88 |
| FLUX + **FreeCure** | **25.75** (+6.63%) | **24.71** (+9.15%) | **24.16** (+10.45%) |

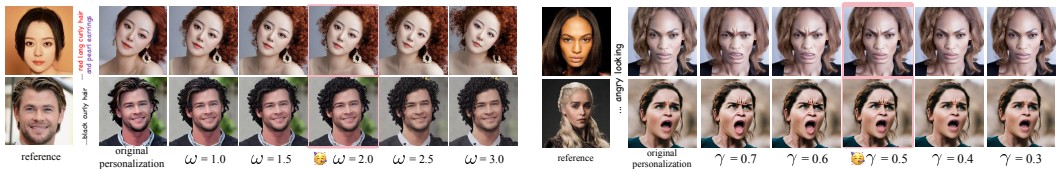

Figure 7: **Performance of FASA w/ and w/o identical initial noises**. FASA can precisely enhance attributes even if PD and FD produce faces with different locations, sizes, and angles.

$Q_p$ × [ $K_p$ , $K_f$ ]     $Q_p$ × [ $K_p$ , $K_f$ ]

Figure 8: **Visualization of the FASA maps** for attribute related area (red points) and non-attribute related area (green points).

hair, sunglasses), the FASA map exhibits greater attention to information from $K_f$, corresponding to FD. Conversely, in regions unrelated to attributes, the FASA map retains its attention on PD. This visualization substantiates the role of FASA in transferring fine-grained attribute information from FD to PD. Additionally, FASA preserves its original attention pattern in regions unrelated to attributes, thereby ensuring the performance of personalization models' identity fidelity. More detailed visualization analyses of FASA are available in Appendix.A.4.6.

## 5.3 Ablation Study

**Overall Analysis of Each Component**. Table 3 presents the individual performance of FASA and APG in enhancing attributes across all baselines. Both components demonstrate positive effects compared to the baseline metrics in Table 1, with FASA showing more noticeable improvements. We attribute this to FASA's ability to effectively handle attributes covering larger areas, such as hair and sunglasses, resulting in more observable enhancements.

(a) Ablation of FASA scaling mask strategy.    (b) Ablation of APG intermediate timesteps.

Figure 9: Ablation Studies.

**Scaling Mask Strategy**. Figure 9a illustrates the evaluation of the scaling mask strategy implemented in FASA. Low scaling values critically hinder FASA's capacity to effectively transfer attribute information to the personalization denoising. On the contrary, high scaling factors may negatively impact the overall quality of the generated faces. Generally, the optimal value of $\omega$ should be around 2.0. More ablation studies of FASA are available in Appendix.A.4.5.

**Effect of Inversion's Intermediate Timesteps**. Figure 9b demonstrates the effects of different starting timesteps in APG, represented by the parameter $\gamma$ in Section 4.2. To achieve a noticeable improvement in attributes while maintaining identity fidelity, an optimal $\gamma$ should be set as 0.5.

Table 3: **Quantitative ablation analysis for FASA and APG's independent effect.** After calculating the metrics for each method, we compute their mean values based on the corresponding foundation model type. For metrics of each baseline, please refer to Appendix.A.4.4.

| Foundation Model | FASA | APG | PC(%) ↑ | IF(%) ↑ | PC × IF (hMean) ↑ |
|---|---|---|---|---|---|
| SDv1.5 | ✓ | ✗ | 21.83 | 46.39 | 29.69 |
|  | ✗ | ✓ | 20.58 | 46.81 | 28.59 |
| SDXL | ✓ | ✗ | 24.55 | 56.97 | 34.31 |
|  | ✗ | ✓ | 23.97 | 56.87 | 33.73 |
| FLUX | ✓ | ✗ | 24.50 | 75.66 | 37.01 |
|  | ✗ | ✓ | 24.09 | 75.85 | 36.57 |

**Hyperparameter Analysis**. Our supplemental ablation analysis indicates that the hyperparameters $\omega$ and $\gamma$ consistently fall within specific ranges across different baselines: $\omega \in [1.8, 2.4], \gamma \in [0.5, 0.6]$. Baselines based on Stable Diffusion v1.5 [49, 64, 62] require larger $\omega$ values (up to 2.4), while other methods [16, 18, 27, 59] perform well around $\omega = 2.0$. Since results remain stable within these ranges, we adopt a unified setting ($\omega = 2.0, \gamma = 0.5$) throughout our experiments.

## 5.4 User Study

To further validate FreeCure's superiority, we conducted an online user study with 30 participants. The study was designed as follows: for each baseline method, we randomly selected 10 samples. Each sample included a reference image, a text prompt, the baseline's output, and the result refined by FreeCure. Participants were asked to evaluate the prompt consistency and identity fidelity of each output. As shown in Fig.10a and Fig.10b, the results demonstrate a clear preference for FreeCure on prompt consistency and equal preference on identity fidelity, indicating that FreeCure's potential negative impact on identity preservation is minimal.

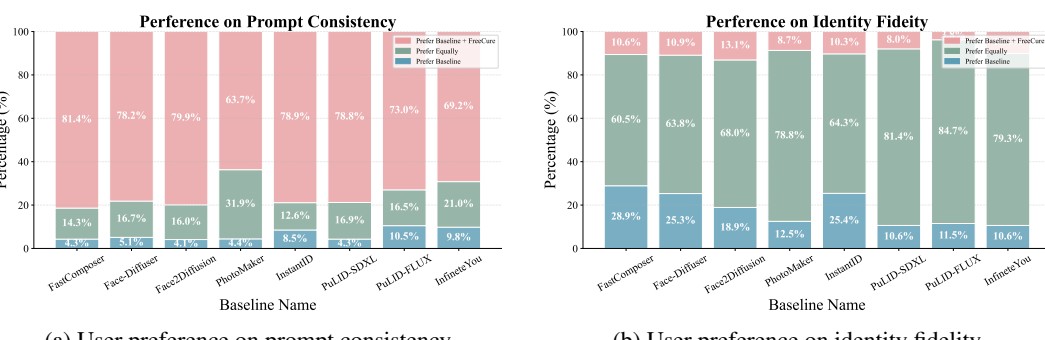

(a) User preference on prompt consistency     (b) User preference on identity fidelity

Figure 10: **User study of FreeCure**. The preference ratio indicate that FreeCure can improve prompt consistency without undermining identity fidelity of different personalization models.

## 6 Limitation and Conclusion

**Limitation.** FreeCure is influenced by inherent bias and maximum achievable capabilities of personalization models, which is commonly observed in plug-and-play adaptation frameworks. Furthermore, it struggles with certain transparent objects, such as glass bottles. In some cases, we also observe slight attribute entanglement between FD and PD processes. Furthermore, FreeCure's application to personalization models based on auto-regressive architectures could be further explored.

**Conclusion.** Facing the challenge that personalization models employing identity embeddings frequently struggle to preserve prompt consistency, we introduce FreeCure, a training-free framework that leverages the high prompt consistency inherent in foundation models to refine the output of personalization models, leading to a remarkable improvement in their prompt consistency and minimal disruption to their original identity fidelity Our experiments validate the effectiveness of FreeCure on popular baselines from different foundation models, particularly in scenarios with complex prompts that encompass multiple attributes.

## Acknowledgments

The research was supported in part by Theme-based Research Scheme (T45-205/21-N) from Hong Kong RGC, Generative AI Research, Development Centre from InnoHK, in part by the National Natural Science Foundation of China (Grant No. 62372480), in part by HKUST-MetaX Joint Lab Fund (No. METAX24EG01-D). We also thank Yubai Wei, Moheng Li, and Meixin Zhou for their assistance with the user study.

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

# A  Appendix

The Appendix consists of three sections:

- Sec.A.1 includes an ethics statement that addresses the privacy of our evaluation datasets and our methods for mitigating potential risks.

- Sec.A.2 provides implementation details, including details of setting up the dual denoising paradigm for every baseline. Details of segmentation models, prompt datasets as well as evaluation metrics are also included.

- Sec.A.3 provides additional results of FreeCure in a boarder range of references, including non-celebrities. It also includes complementary FD outputs corresponding to the results of the main submission (Fig.5, 6), and broader attribute enhancement achieved through the integration of advanced models, such as Segment-Anything Model [28].

- Sec.A.4 provides a comprehensive analysis of FreeCure, including analysis for initial noise conditions, detailed visualization of FASA modules, more detailed ablation studies, runtime analysis and a template of the user study.

## A.1  Ethics Statements

We confirm that our human face datasets, featuring both celebrities and non-celebrities, are sourced from public datasets or Google Images, limited to CC-licensed content. We have made extensive efforts to balance critical attributes like gender, race, and age to mitigate significant bias. Furthermore, we acknowledge that FreeCure could be misused for malicious purposes, such as creating impersonating identities. To mitigate these risks, we are committed to ethical governance and a controlled release strategy. For instance, we release the method only to accredited researchers under a license prohibiting misuse, with mandatory ethics training.

## A.2  More Implementation Details

### A.2.1  Triggering Hidden Foundation Knowledge in Personalization Baselines

The core architecture of FreeCure is based on a dual denoising paradigm of the FD and PD process, which triggers hidden foundation knowledge in personalization models. This paradigm deactivates the identity embeddings which are integrated into the cross-attention layers. In Table 4 we illustrate how we deactivate identity embedding for different baselines mentioned in the main paper since they have various identity embedding fusion strategies (e.g., LoRA, adapters) and prompt templates.

Table 4: **Strategies for implementing dual denoising paradigms on different baselines.** "black curly hair" is used here as an example and can be replaced with other facial attributes.

| Method | PD inputs | FD inputs |
|---|---|---|
| FastComposer | a \<P> \<\|image\|> with black curly hair | a \<P> image with black curly hair |
| Face-Diffuser | a \<P> \<\|image\|> with black curly hair | a \<P> image with black curly hair |
| Face2Diffusion | a f l with black curly hair | a \<P> image with black curly hair |
| InstantID | a \<P> with black curly hair | a \<P> image with black curly hair |
| PhotoMaker | a \<P> img with black curly hair | a \<P> image with black curly hair |
| PuLID | a \<P> with black curly hair | a \<P> image with black curly hair |
| InfiniteYou | a \<P> with black curly hair | a \<P> image with black curly hair |

FastComposer [64], Face-Diffuser [62] and PhotoMaker [34]share the similar prompting template, wherein their identity embeddings are integrated into the "\<P>" word, which is initialized with class-specific words (*e.g., man, woman, boy, girl*). To establish the FD process, we maintain the original plain text embedding for "\<P>" without incorporating any identity information, effectively equivalent to applying a null identity embedding. Face2Diffusion [49] integrated the identity embedding into the trigger word "f". Therefore, we replace the identity embedding with a zero-valued tensor before its

Table 5: **Information for facial attribute and label relationship of BiSeNet.**

| label | attribute | label | attribute | label | attribute |
|---|---|---|---|---|---|
| 0 | background | 6 | glasses | 12 | lower lip |
| 1 | facial skin | 7 | right ear | 13 | neck |
| 2 | right eyebrow | 8 | left ear | 14 | necklace |
| 3 | left eyebrow | 9 | nose | 15 | cloth |
| 4 | left eye | 10 | earrings | 16 | hair |
| 5 | right eye | 11 | upper lip | 17 | hat |

Table 6: **Prompts for evaluation categorized by the number of included attributes (range from 1 to 3).** `` will be replaced with placeholder tokens such as `man`, `woman`, `boy`, etc.

| Attribute | Prompt |
|---|---|
| 1 | a `` with black curly hair
a `` with blonde curly hair
a `` with red long straight hair
a `` with very angry looking
a `` with frowning worriedly
a `` laughing happily
a `` wearing a mask
a `` wearing sunglasses |
| 2 | a `` with white curly hair, frowning worriedly
a `` with black curly hair, laughing happily
a `` with blonde curly hair and blue eyes
a `` with blue eyes, laughing happily
a `` wearing sunglasses, laughing happily
a `` with black hair, wearing silver earrings
a `` with blonde hair and blue eyes
a `` with sunglasses, frowning worriedly |
| 3 | a `` with red curly hair, wearing pearl earrings, unhappy looking
a `` with blue eyes and blonde curly hair, smiling
a `` with white curly hair, wearing sunglasses, laughing happily
a `` with black curly hair, wearing silver earrings, frowning worriedly |

merging process. For baselines such as InstantID [59], PuLID [18], and InfiniteYou [27] that employ residual cross-attention adapters for identity fusion, we initialize their identity embeddings as zero tensors with matching dimensionality while preserving the original textual embeddings unchanged.

### A.2.2 Attribute Segmentation Models

As mentioned in Sec.5, we use two mainstream segmentation models: BiSeNet and Segment-Anything models for attribute-aware mask generation. BiSeNet [2] is a popular framework for face parsing that is capable of generating semantic masks corresponding to facial attributes. Table 5 shows BiSeNet's prediction label and corresponding facial attributes, demonstrating that it can address the majority of facial attributes, thereby verifying the robustness of FreeCure. Additionally, FreeCure is designed to be seamlessly integrated with various segmentation models, not limited to BiSeNet. We have showcased results that utilize Segment-Anything as the face parsing model in Sec.A.3.4.

### A.2.3 Facial Prompts for Evaluation

Table 6 introduces the prompts for evaluating enhancement performance for facial attributes. Generally, our prompts include multiple facial attributes, ensuring that previous baselines' weaknesses in prompt consistency can be fully uncovered and highlighting the enhanced performance of FreeCure.

---

[2]https://github.com/CoinCheung/BiSeNet

### A.2.4 Details of Metrics

**Prompt Consistency (PC, also known as CLIP-T).** We leverage the official implementation of the vision transformer[3] provided by OpenAI.

**Identity Fidelity (IF).** We use the official implementation of MTCNN + FaceNet pipeline[4] to conduct the processes of face detection and feature extraction from facial regions. Additionally, we compute the cosine similarity between two face embeddings to evaluate their similarity.

**Face Diversity (Face Div.).** We utilize the official implementation of LPIPS[5] to quantify the perceptual distance between two facial images.

### A.3 More Results

### A.3.1 More Results on Celebrity & Non-Celebrity References

In Fig.11, 12 and 13, we provide additional experimental results on more reference images to further validate the consistent performance of FreeCure. This robust performance shows the potential of FreeCure to enhance various personalization models in practical applications, as the ability to handle non-celebrity personalization is a critical requirement in real-world scenarios.

---

[3]https://github.com/openai/CLIP
[4]https://github.com/timesler/facenet-pytorch
[5]https://github.com/richzhang/PerceptualSimilarity

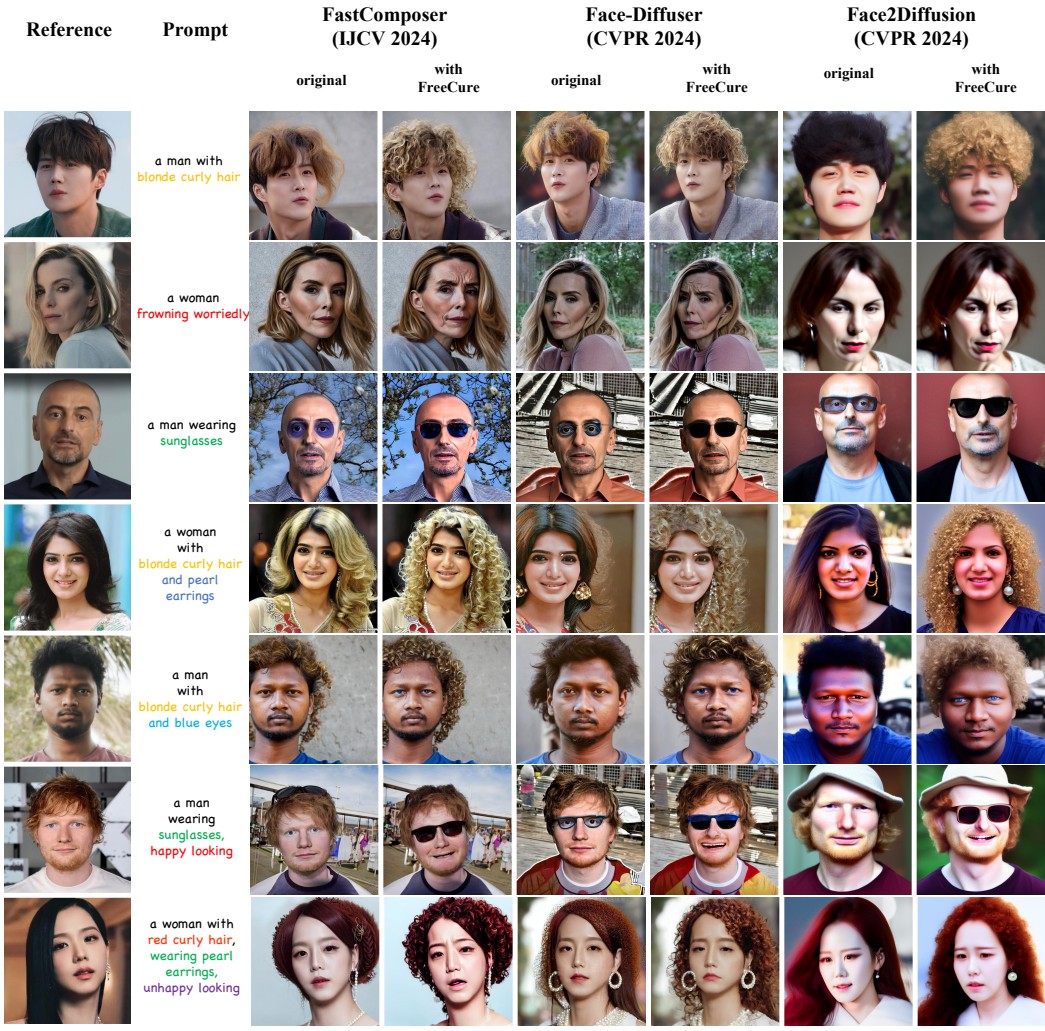

Figure 11: **More results for baselines implemented with SDv1.5, including FastComposer, Face-Diffuser and Face2Diffusion.** We provide personalization results including non-celebrity identities.

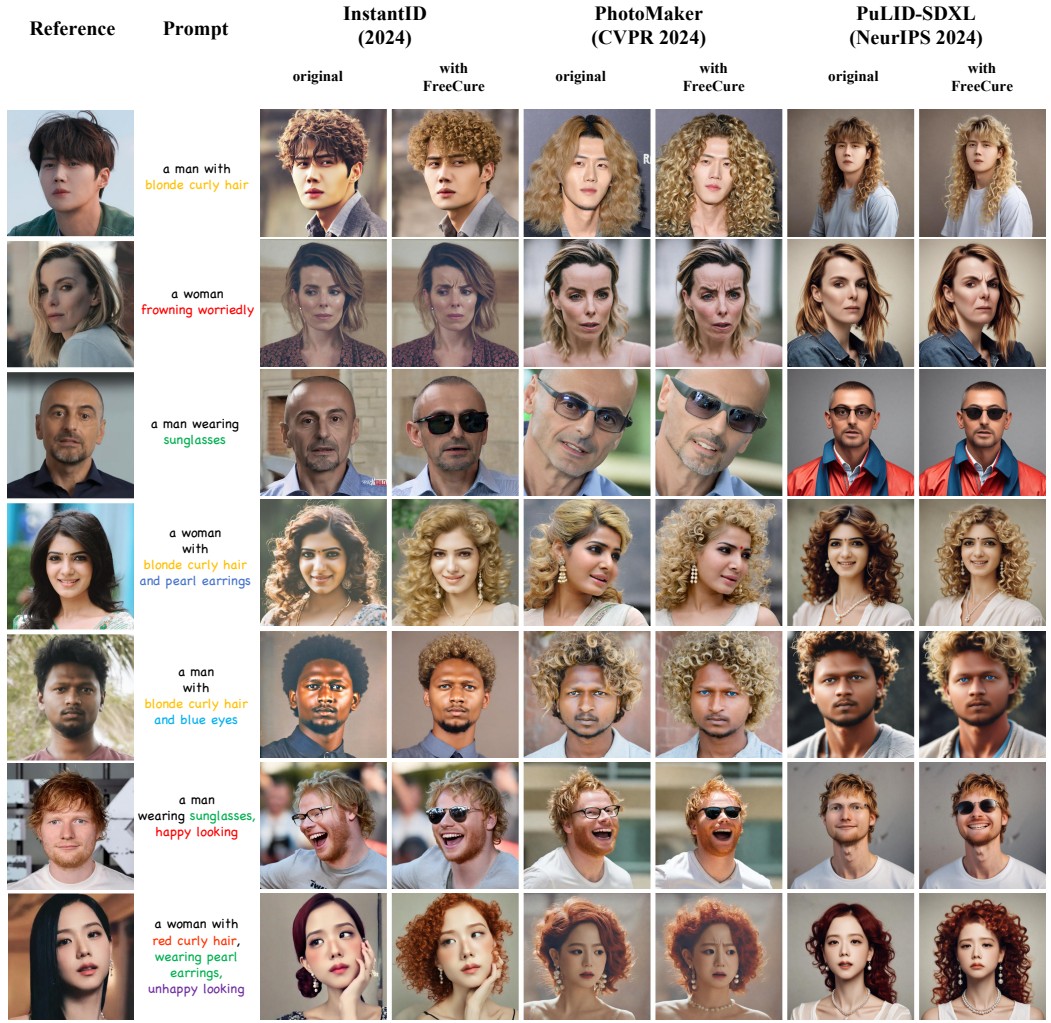

Figure 12: **More results for baselines implemented with SDXL, including InstantID, PhotoMaker and PuLID-SDXL.** We provide personalization results including non-celebrity identities.

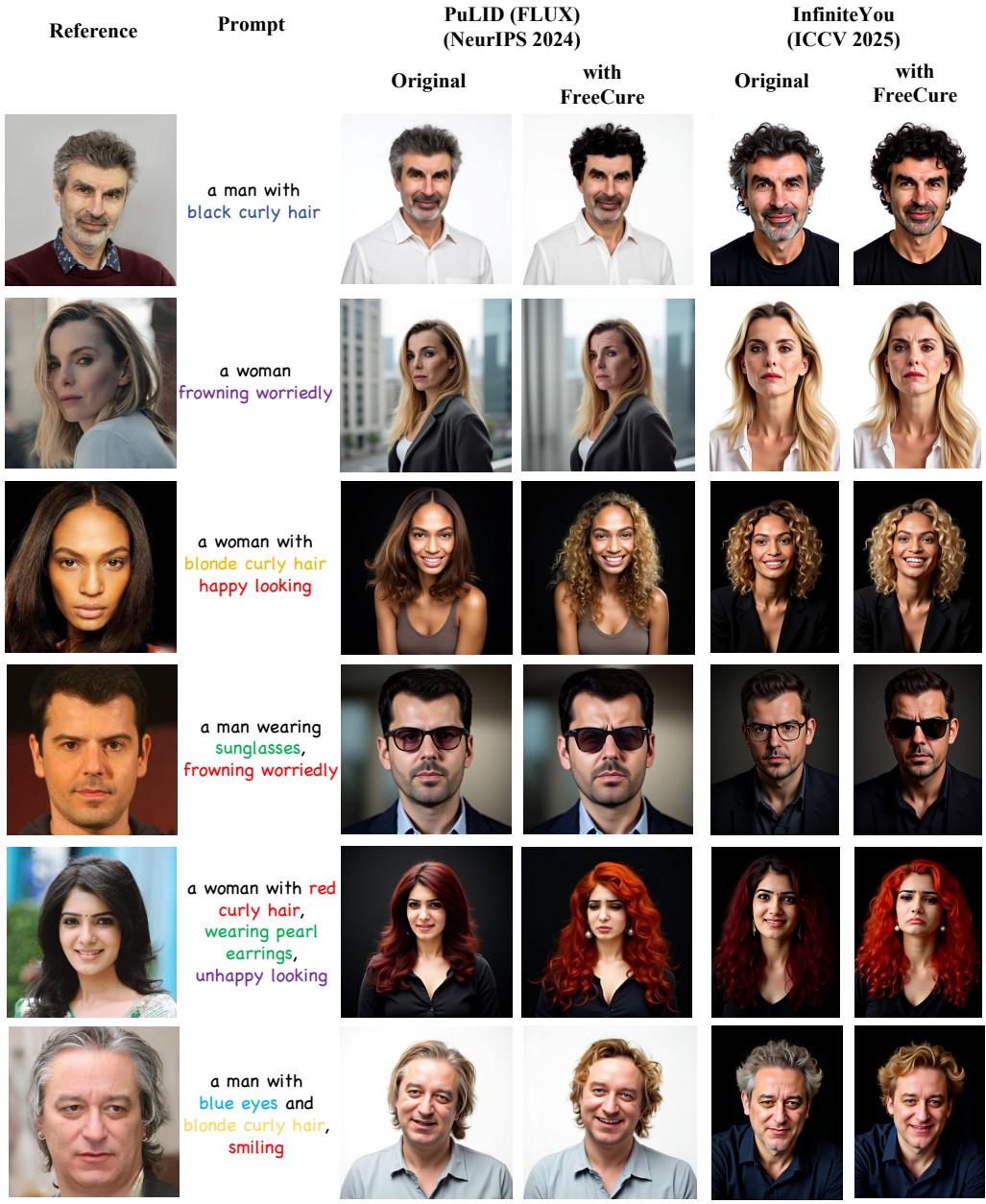

Figure 13: **More results for baselines implemented with FLUX, including PuLID-FLUX and InfiniteYou.** We provide personalization results including non-celebrity identities.

### A.3.2 Supplementary Comparison of Results for Main Paper

To better demonstrate the efficacy of FreeCure, Fig.14, 15 present a comparative analysis of the results generated by the FD, PD, and enhanced outputs by FreeCure linked to Fig.5. The results indicate that FreeCure faithfully transfers target prompt-aligned attributes from the FD into the PD processes while preserving the original identity information.

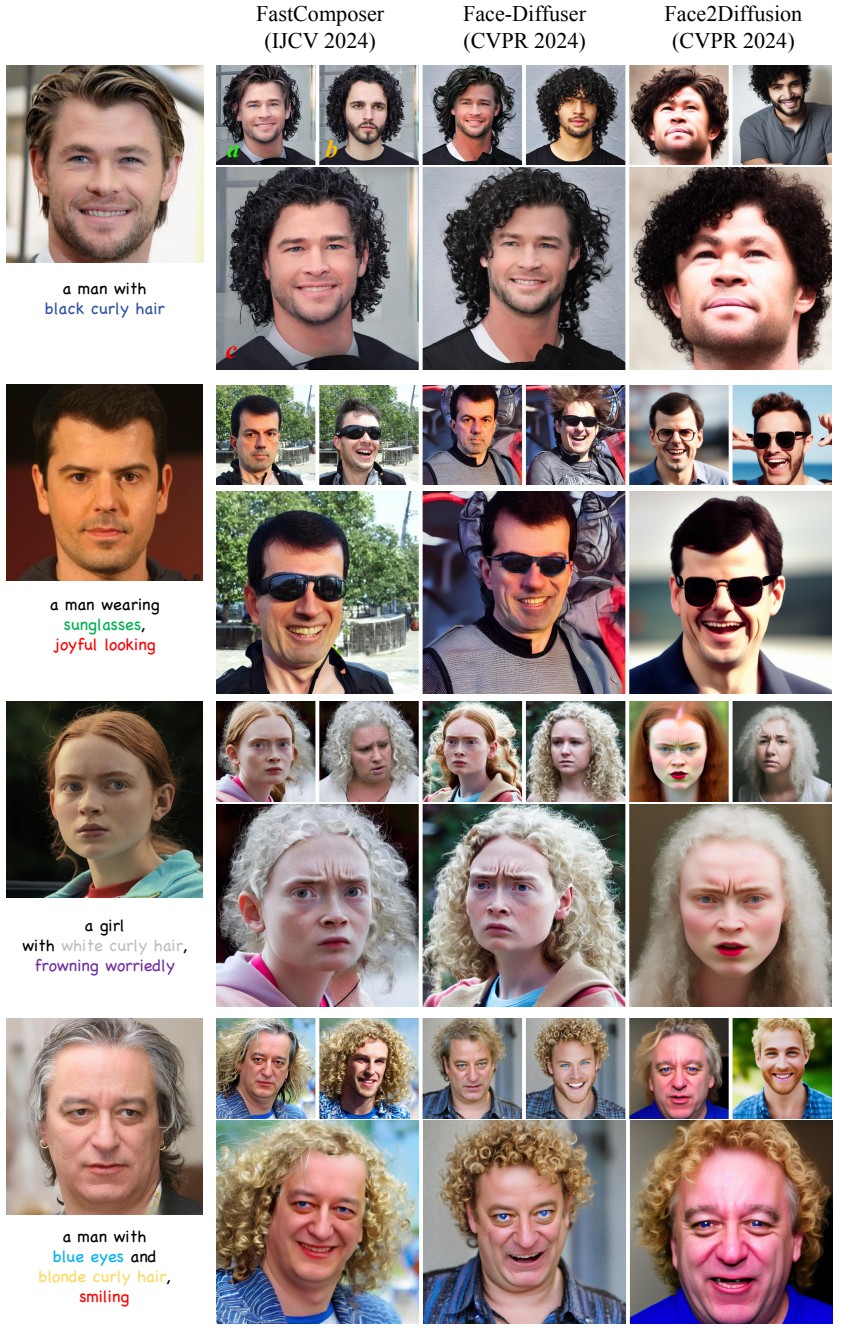

Figure 14: **Corresponding comparative FD & PD outputs for results in the main submission. (FastComposer, Face-Diffuser and FaceDiffusion)** (a) refers to original personalization outputs, (b) refers to foundation outputs, and (c) refers to outputs enhanced with proposed FreeCure.

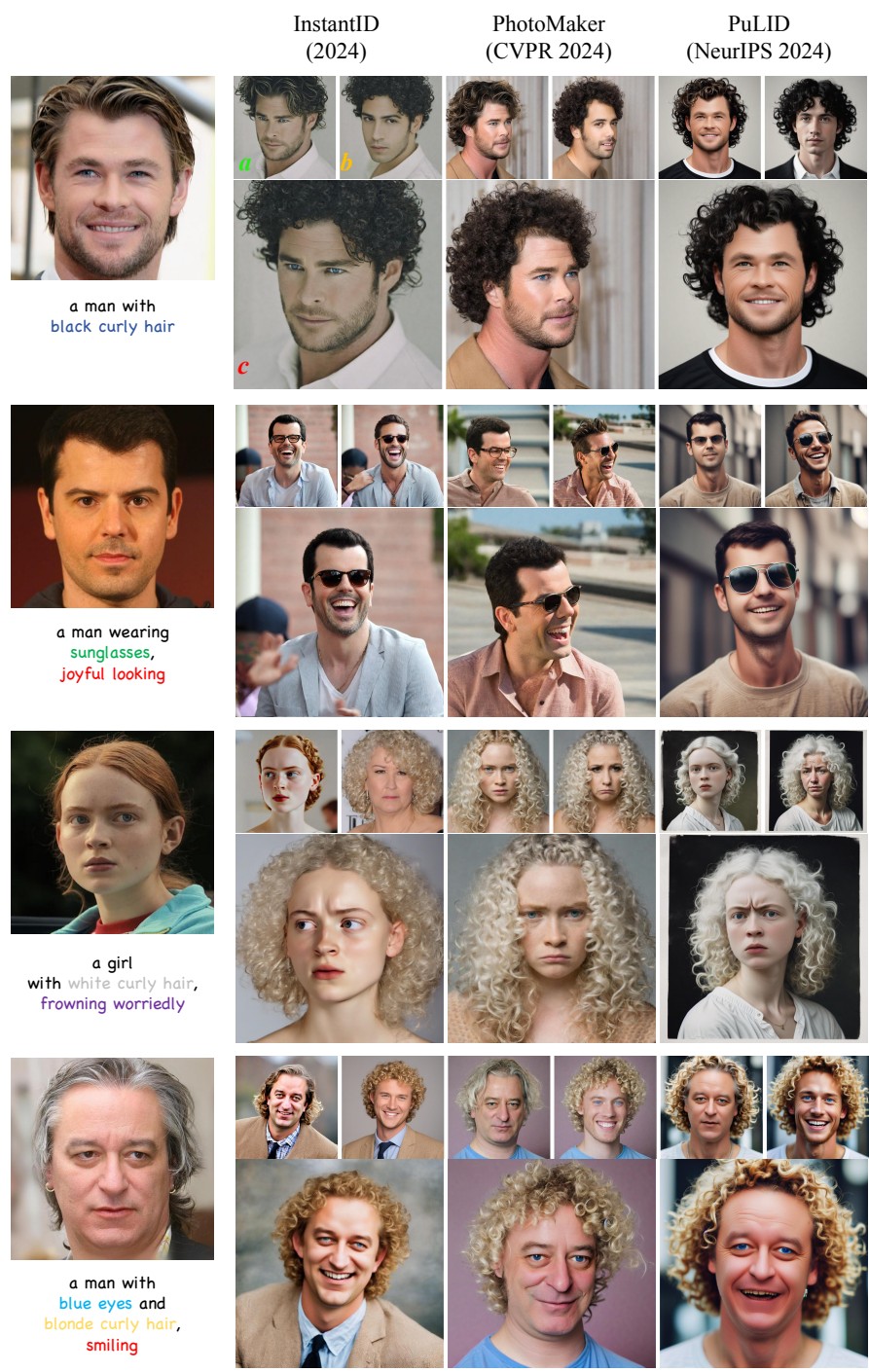

Figure 15: **Corresponding comparative FD & PD outputs for results in the main paper. (InsantID, PhotoMaker and PuLID)** (a) refers to original personalization outputs, (b) refers to foundation outputs, and (c) refers to outputs enhanced with proposed FreeCure.

### A.3.3    More Detailed Quantitative Results for Prompt Consistency with Multiple Attributes.

As supplementary results for Table.2, we provide source prompt consistency results of each baseline considering prompts with various attributes in Table.7. It shows that as the prompt becomes increasingly complicated, all baselines are facing more decreasing in prompt consistency, which can limit their application in real scenarios. The results also support the fact that FreeCure's improvement in PC becomes more significant as the complexity of the prompt increases.

Table 7: Quantitative comparison of prompt consistency with different number of attributes.

| Method | PC (1 Attr.) ↑ | PC (2 Attr.) ↑ | PC (3 Attr.) ↑ |
|---|---|---|---|
| FastComposer | 18.83 | 18.05 | 16.92 |
| FastComposer + **FreeCure** | **21.42** (+13.75%) | **21.29** (+17.95%) | **19.69** (+16.37%) |
| Face-Diffuser | 21.68 | 20.98 | 18.02 |
| Face-Diffuser + **FreeCure** | **22.89** (+5.58%) | **22.72** (+8.29%) | **21.17** (+17.48%) |
| Face2Diffusion | 22.54 | 21.98 | 20.54 |
| Face2Diffusion + **FreeCure** | **23.81** (+5.63%) | **23.02** (+4.73%) | **22.63** (+10.18%) |
| InstantID | 22.43 | 21.81 | 20.95 |
| InstantID + **FreeCure** | **23.98** (+6.91%) | **23.52** (+7.84%) | **23.11** (+10.31%) |
| PhotoMaker | 23.51 | 23.01 | 22.57 |
| PhotoMaker + **FreeCure** | **25.04** (+6.51%) | **24.91** (+8.26%) | **24.64** (+11.19%) |
| PuLID (SDXL) | 25.56 | 25.12 | 24.43 |
| PuLID (SDXL) + **FreeCure** | **26.30** (+3.34%) | **25.96** (+5.32%) | **25.73** (+3.55%) |
| PuLID (FLUX) | 23.29 | 22.08 | 21.32 |
| PuLID (FLUX) + **FreeCure** | **25.52** (+9.57%) | **24.49** (+10.91%) | **23.87** (+11.96%) |
| InfiniteYou | 25.01 | 23.19 | 22.43 |
| InfiniteYou + **FreeCure** | **25.98** (+3.88%) | **24.92** (+7.46%) | **24.45** (+9.01%) |

### A.3.4    Integration with Segment-Anything for Attribute Extraction

Fig.16 showcases FreeCure's consistent performance when integrated with more advanced segmentation models such as Segment-Anything. For instance, the attribute **mask** cannot be handled by BiSeNet since its label is absent. However, by replacing BiSeNet with Segment-Anything and specifying the target prompt as "mask," valid semantic masks can be accurately generated, thereby guaranteeing effective attribute enhancement through FASA. Given that Segment-Anything extracts attributes based on flexible prompts, its seamless integration significantly enhances the versatility of FreeCure, enabling it to address a broader range of scenarios.

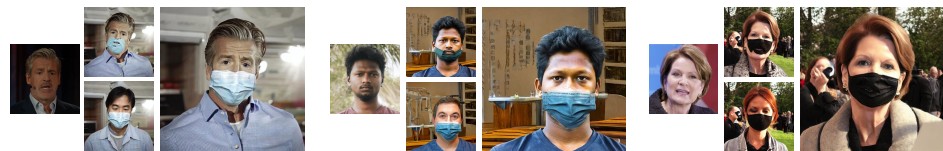

Figure 16: **Results for FreeCure's integration with Segment-Anything.** FreeCure can seamlessly integrate with more advanced segmentation models such as Segment-Anything for extraction of a boarder range of attributes. Target prompt: "a person wearing a mask".

### A.3.5    Analysis on non-facial attributes enhancement

We have conducted supplementary experiments about attributes beyond facial elements. Following the prompt set of PuLID, we evaluated prompts involving common objects, clothing, and other typical non-facial attributes, using Segment Anything for mask generation. The experimental results across several baselines are shown in Table.8:

Table 8: Quantitative comparison of on non-facial attributes.

| | PC(%) ↑ | IF(%) ↑ | Face Div. (%) ↑ |
|---|---|---|---|
| Face2Diffusion | 22.87 | 41.08 | 41.34 |
| Face2Diffusion + FreeCure | 23.34 (+2.06%) | 40.75 (-0.80%) | 41.91 (+1.38%) |
| InstantID | 22.43 | 65.55 | 47.73 |
| InstantID + FreeCure | 23.51 (+4.81%) | 65.34 (-0.32%) | 48.06 (+0.69%) |
| PhotoMaker | 24.67 | 52.14 | 46.02 |
| PhotoMaker + FreeCure | 25.37 (+2.84%) | 51.77 (-0.71%) | 46.58 (+1.21%) |
| PuLID (SDXL) | 27.43 | 59.41 | 41.97 |
| PuLID (SDXL) + FreeCure | 28.13 (+2.55%) | 59.03 (-0.64%) | 42.05 (+0.19%) |

Experimental results indicate that the facial personalization model exhibits a weaker copy-paste effect in non-facial regions, leading to less pronounced improvements in prompt consistency compared to the facial attributes specifically addressed in our submission. However, we observed that FreeCure still demonstrates significant restoration effects for attributes such as headphones and hats, which are in close proximity to the face (see Fig.17). Moreover, since the restoration occurs outside the facial region, FreeCure has a more negligible negative impact on identity fidelity.

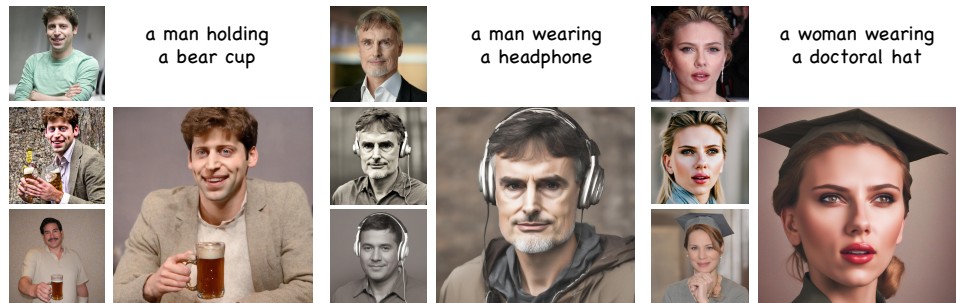

Figure 17: **Results on non-facial attributes.** FreeCure can still improve prompt consistency on non-facial attributes, especially on objects whose location is close to face.

### A.4 More Analysis

#### A.4.1 Non-interference Enhancement of Multiple Facial Attributes

We validate that the subsequent enhancement processes of FreeCure do not impact attributes that have already been enhanced. Fig.18 visualizes the intermediate results at FreeCure's different stages. When APG is used in the latter stages, attributes previously enhanced through FASA remain unaffected. This illustrates that FreeCure's improvement across multiple attributes is highly robust and consistent.

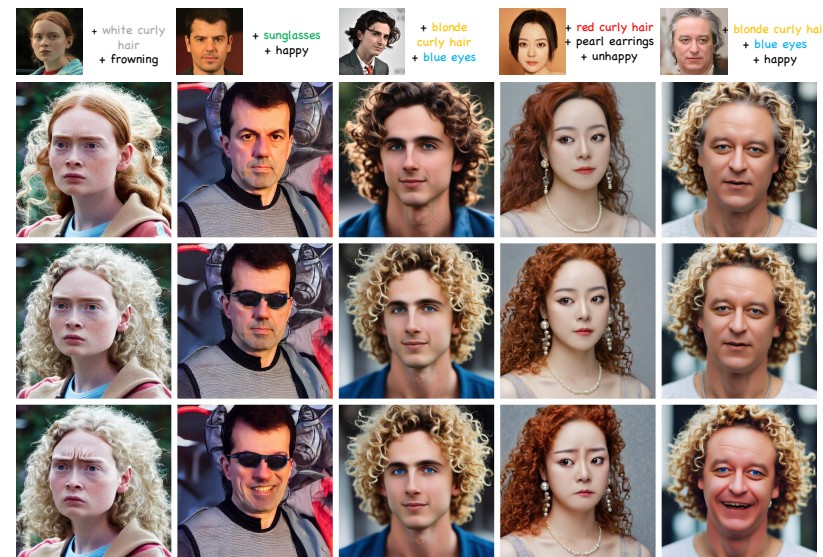

Figure 18: **Demonstration of FreeCure's non-interference manner**. **top row**: personalization models' outputs; **middle row**: intermediate enhanced results with FASA; **bottom row**: final results after APG enhancement.

#### A.4.2 More Robustness Justification under Inaccurate Attribute Masks

Although FreeCure rely on attribute segmentation process based on pretrained semantic segmentation models. Its performance does not sensitive to the accuracy of these models. We design two-way validation experiments that introduce inaccuracy to attribute masks. First, we apply a dilation operation on masks, which might include area which does not belong to the target attribute area. Second, we apply an erosion operation, which might cause loss of some useful information about attributes. To create significant distortion of masks, we set the receptive field of both dilation and erosion to 25 pixels. We conduct experiments on several baselines, and the Table.9 below reports the results for respective prompt consistency (PC) and identity fidelity (IF).

Table 9: PC and IF Performance for Inaccurate Masks

|            | original IF | original PC | dialated IF | dialated PC | eroded IF | eroded PC |
|------------|-------------|-------------|-------------|-------------|-----------|-----------|
| InstantID  | 62.01       | 23.62       | 62.23       | 23.45       | 62.23     | 23.21     |
| PhotoMaker | 50.15       | 24.91       | 50.01       | 24.87       | 50.44     | 24.34     |
| PuLID-SDXL | 56.95       | 26.05       | 56.74       | 25.51       | 57.01     | 25.36     |
| PuLID-FLUX | 72.61       | 24.78       | 72.39       | 24.52       | 72.97     | 24.32     |

The results demonstrate that identity fidelity (IF) remains largely consistent across both dilated and eroded mask conditions, with slight improvements observed in eroded masks. This occurs because mask erosion preserves certain attributes similar to the reference image. As for prompt consistency, both dilation and erosion cause minimal degradation, though erosion exhibits marginally greater

impact. We consider that this stems from cases where fine-grained attribute restoration (e.g., iris color, earrings) may fail when the mask completely degenerates due to erosion.

Generally, the performance degradation of FreeCure due to inaccurate masks is limited. Moreover, in practical applications, specialized segmentation models (e.g., BiSeNet, Segment Anything) exhibit high robustness, making failure cases caused by mask inaccuracies even rarer than observed in these deliberately designed experiments. This further validates the inherent robustness of the FreeCure framework.

### A.4.3 More Comparison Results w/ and w/o Identical Initial Noise

We conduct extensive experiments to evaluate the impact of initial noise conditions. Fig.19 demonstrates that under identical initial noise conditions, FreeCure consistently enhances target attributes across different input reference images. This finding highlights the robustness and practical applicability of FreeCure. Fig.20 illustrates that FreeCure can successfully inject attribute information from the FD process to the PD process even if their final generated faces have various locations, angles, and sizes. This finding particularly demonstrates the robustness of the FASA module.

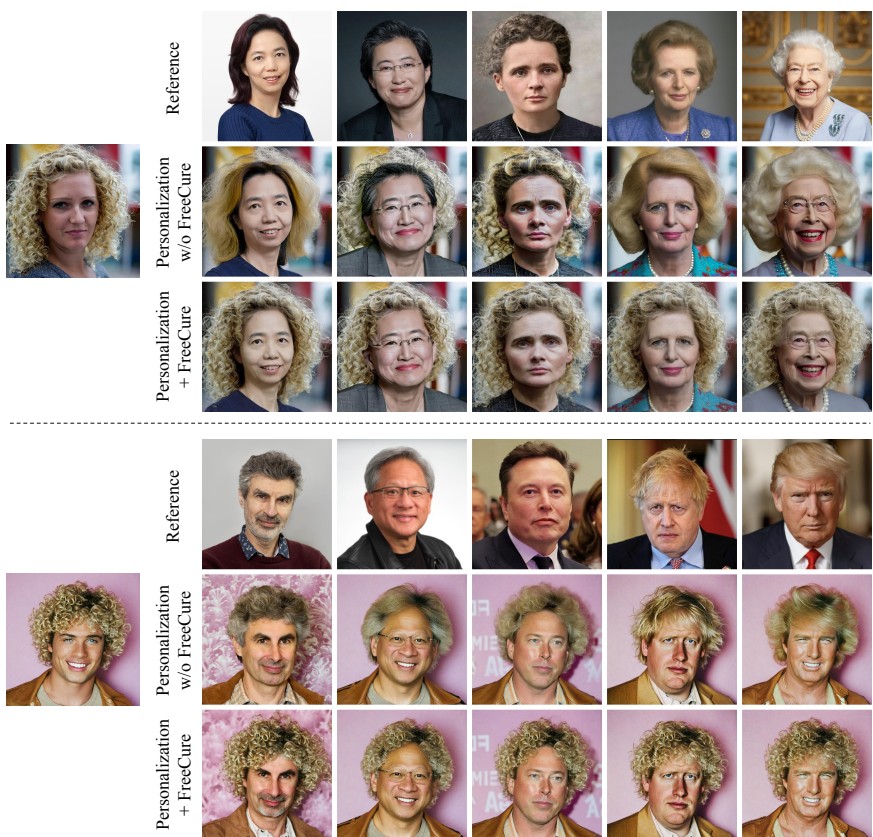

Prompt: a <person> with blonde curly hair

Figure 19: **More studies for attribute enhancement start with identical initial noise.** FreeCure exhibits robust performance with the single foundation output's guidance on personalized outputs from various references (generated based on Face-Diffuser and PhotoMaker).

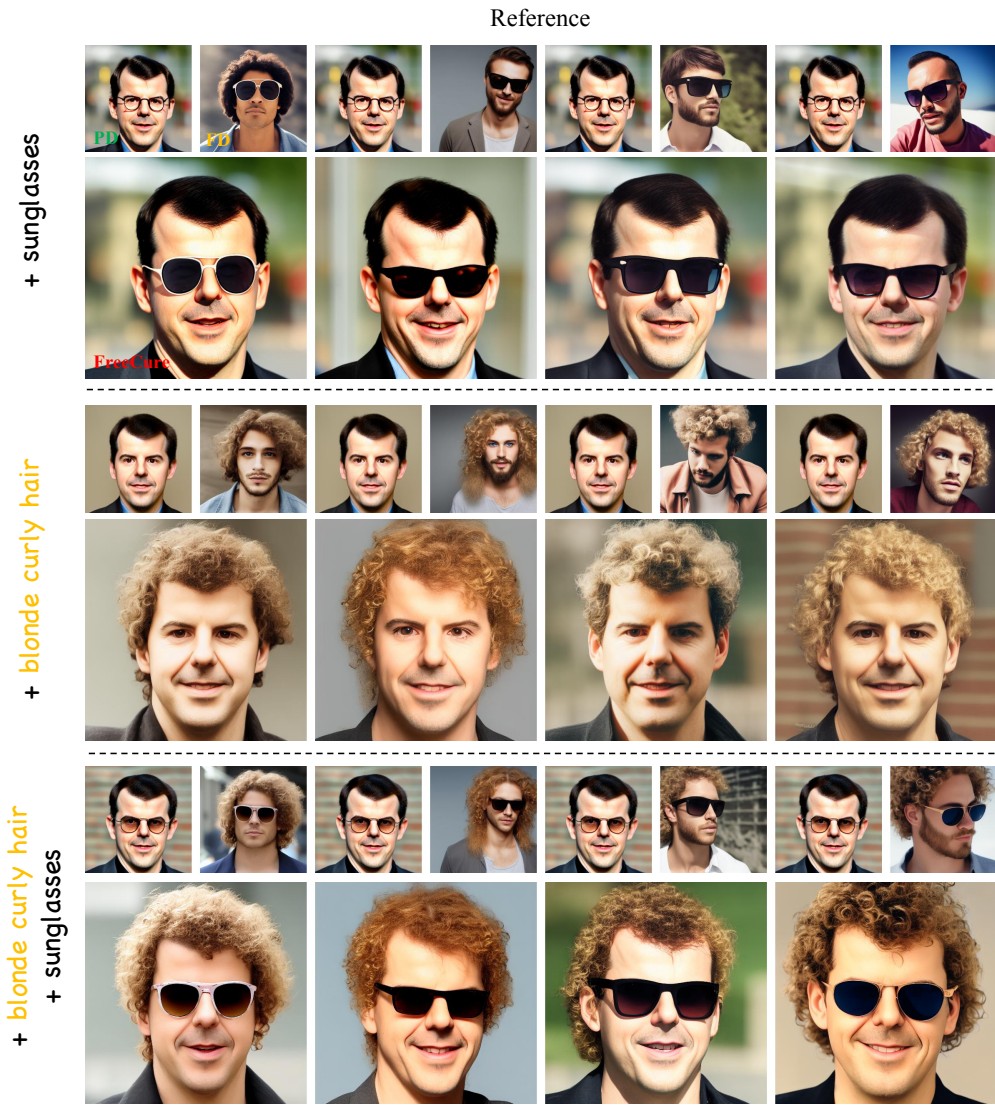

Figure 20: **More validation on FreeCure's robust performance with different initial noises for PD and FD processes**. Even if PD and FD's outputting faces have totally different locations, angles, and sizes, FreeCure can still exhibit stable enhancement performance.

### A.4.4 More Detailed Quantitative Results for Ablation Studies

Table.10 provides a supplementary details for ablation studies mentioned in Table.3. The results indicate that both FASA and APG consistently improve prompt consistency across all original personalization models. Notably, FASA yields more substantial improvements compared to APG, aligning with our findings in Sec.5.3.

Table 10: **Quantitative ablation analysis for FASA and APG's independent effect on overall performance.** The table provides detailed ablation studies on each baselines.

| Method | FASA | APG | PC(%) ↑ | IF(%) ↑ | PC × IF (hMean) ↑ |
|---|---|---|---|---|---|
| FastComposer | ✓ | ✗ | 20.91 | 41.94 | 27.91 |
|  | ✗ | ✓ | 19.03 | 42.84 | 26.35 |
| Face-Diffuser | ✓ | ✗ | 21.86 | 57.92 | 31.74 |
|  | ✗ | ✓ | 20.74 | 58.11 | 30.60 |
| Face2Diffusion | ✓ | ✗ | 22.73 | 39.30 | 28.80 |
|  | ✗ | ✓ | 21.98 | 39.47 | 28.24 |
| InstantID | ✓ | ✗ | 23.02 | 62.45 | 33.64 |
|  | ✗ | ✓ | 22.76 | 62.51 | 33.40 |
| PhotoMaker | ✓ | ✗ | 24.76 | 50.62 | 33.25 |
|  | ✗ | ✓ | 23.61 | 50.86 | 32.25 |
| PuLID (SDXL) | ✓ | ✗ | 25.86 | 57.85 | 35.74 |
|  | ✗ | ✓ | 25.54 | 57.24 | 35.33 |
| PuLID (FLUX) | ✓ | ✗ | 24.12 | 72.98 | 36.26 |
|  | ✗ | ✓ | 23.86 | 73.03 | 35.97 |
| InfiniteYou | ✓ | ✗ | 24.87 | 78.34 | 37.75 |
|  | ✗ | ✓ | 24.32 | 78.67 | 37.15 |

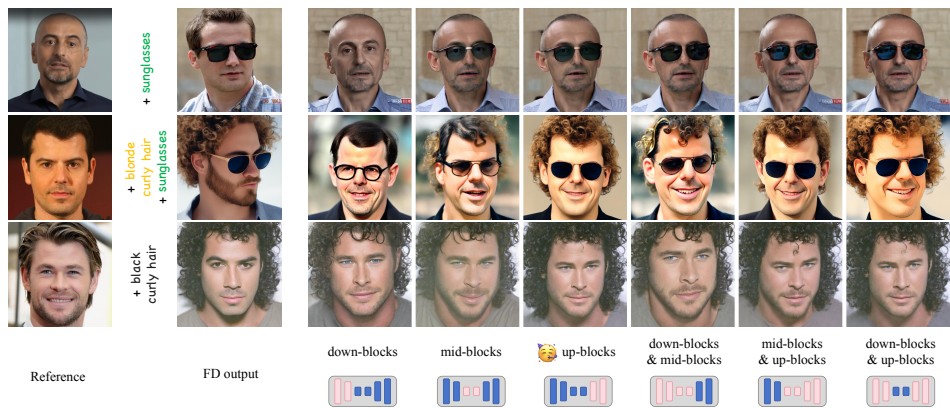

Figure 21: **Ablation study for applying FASA at different positions of UNet**. We find out that applying FASA modules to upsampling blocks alone will be sufficient enough to exhibit promising enhancement for target attributes.

### A.4.5 Ablations of Applying FASA at Different Positions in the Denoising Model

We conduct detailed ablation studies to determine the optimal placement of the FASA module within the denoising model, as illustrated in Fig.21. When FASA is applied exclusively to the downsampling and middle blocks, its effectiveness is limited. In contrast, applying FASA to the upsampling blocks yields the most significant performance improvements. Furthermore, adding FASA to the middle and downsampling blocks provides negligible additional benefits when the upsampling blocks are already

utilized, as the upsampling blocks alone are sufficient to accurately transfer attribute information from the FD to the PD process. Based on these findings, we conclude that applying FASA to the denoising model's upsampling blocks represents the optimal configuration.

### A.4.6 More Visualization of FASA

We provide a more detailed visualization of FASA in Fig.22, which substantiates the claim of Sec. 5.2 that FASA enhances facial attributes in a fine-grained manner. For instance, FASA effectively captures and faithfully transfers information of attributes localized in small areas (e.g. eyes, pearl earrings). Furthermore, in regions unrelated to the target attributes, FASA maintains a strong alignment with the original PD attention maps, demonstrating that it preserves the core functionality of the personalization models.

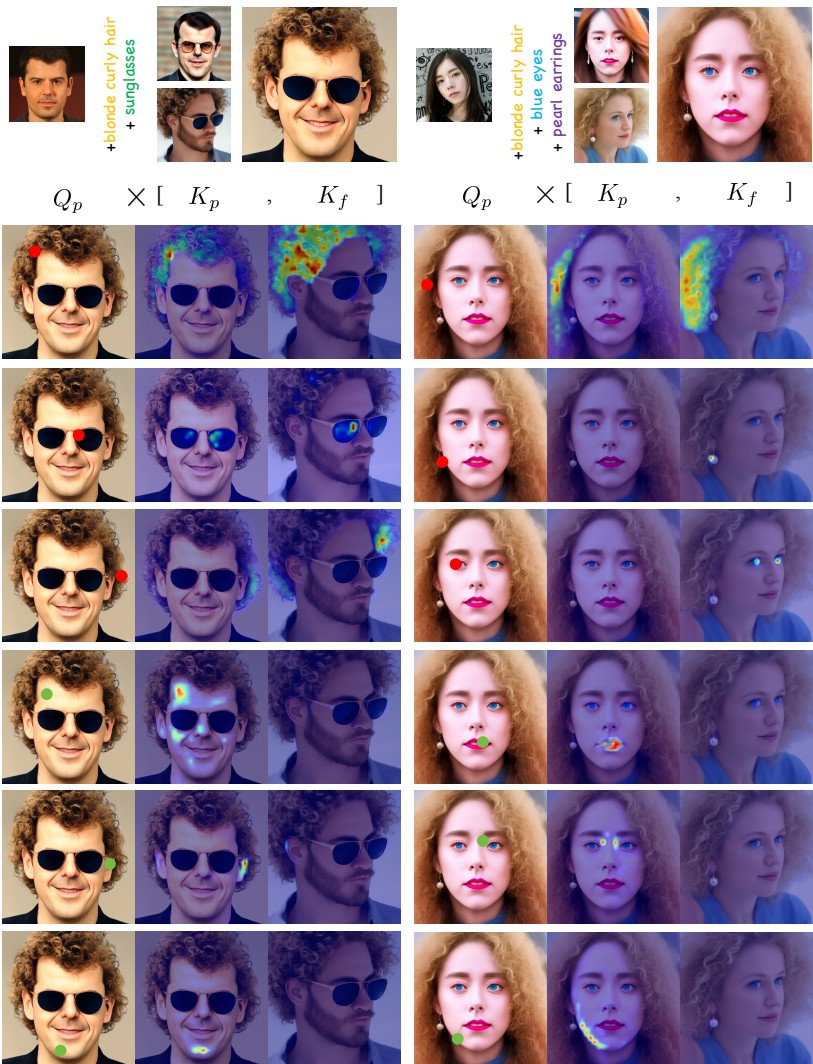

Figure 22: **More visualization of FASA map on attribute-relevant and attribute-irrelevant areas.** Red points refer to the area of target attributes and green points refer to the area which is not related to target attributes.

### A.4.7 More Studies on Identity Embedding

Fig.23 illustrates the impact of identity embedding interpolation when integrated into the cross-attention layers of different blocks, as mentioned in Sec.3. When cross-attention maps of the FD process are injected into the downsampling blocks or middle blocks, the changes in output are minimal, even if all identity embedding's cross-attention maps are replaced ($\alpha = 1$). It is only when the interpolation is applied to the upsampling blocks that significant degradation of identity information is evident, while other facial attributes are effectively restored. In summary, the identity embedding exerts its most significant influence within the upsampling blocks of the denoising model. However, a small value of $\alpha$ can cause significant identity information loss, supporting the argument in Sec.3, that the well-trained cross-attention layers for identity information extraction is susceptible.

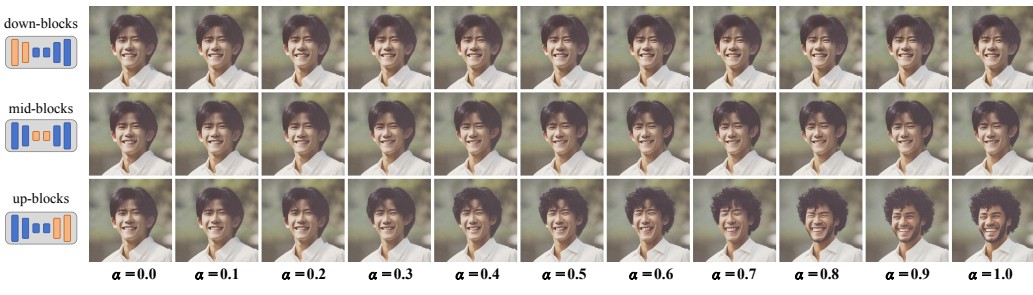

prompt: "a <man> image with black curly hair, laughing happily"

Figure 23: **Identity embedding interpolation's influence on cross-attention layers of different blocks in the denoising model.** It is applied on: downsampling blocks (top), bottleneck blocks (middle), and upsampling blocks (bottom).

We have conducted a detailed analysis using finer-grained $\alpha$ values and their corresponding quantitative outcomes. Below are the results:

Table 11: Fined-grained Quantitative Metrics for Examples of Cross-attention Interpolation in Sec.3

|        |                    | $\alpha$ | 0     | 0.2   | 0.4   | 0.6   | 0.8   | 1     |
|--------|--------------------|------|-------|-------|-------|-------|-------|-------|
| top    | Identity Fidelity  |      | 50.23 | 24.53 | 10.34 | 6.42  | 3.43  | 2.1   |
|        | Prompt Consistency |      | 24.98 | 25.5  | 26.39 | 28.54 | 30.43 | 33.01 |
| bottom | Identity Fidelity  |      | 52.34 | 49.52 | 29.12 | 15.75 | 6.32  | 3.2   |
|        | Prompt Consistency |      | 26.76 | 30.34 | 31.09 | 32.98 | 34.23 | 35.02 |

The results demonstrate that identity fidelity declines sharply even when only a small portion of the original cross-attention map is altered, suggesting that identifying *a well-optimized sweet spot* through cross-attention manipulation alone is highly challenging. In summary, difficulties in cross-attention manipulation provide us with a more solid motivation in the exploration of self-attention modules.

### A.4.8 Inference Time Analysis

We conduct experiments on a single H800 GPU with 80 GB VRAM to measure each baseline's inference time before and after applying FreeCure. Table.12 below reports the results ("*" means 30-step denoising due to official guidance and 50-step denoising is applied for the rest baselines)

It is true that introducing extra calculation can lead to longer runtime, we would like to emphasize that most training-free methods introduce extra processes, including inversion operation, denoising processes, and extra attention calculation. Plus, comparing to the fact that most encoder-based personalization methods require a long range time of data collection, preprocessing and tuning (this may consume a large array of GPUs and thousands of GPU-hours), our training-free method provides an innovative perspective for prompt consistency improvement by the knowledge in themselves, without the need for designing new model architecture and dataset curation.

Table 12: Runtime analysis

| | Baseline (seconds) | Baseline + FreeCure (seconds) | | | |
|---|---|---|---|---|---|
| | | Stage 1 | Stage 2 with FASA | APG | Total |
| FastComposer | 1.79 | 1.96 | 2.19 | 1.83 | 5.98 |
| Face-Diffuser | 2.13 | 2.63 | 2.33 | 1.92 | 6.88 |
| Face2Diffusion* | 1.13 | 1.25 | 1.47 | 1.23 | 3.95 |
| PhotoMaker | 6.52 | 7.04 | 10.03 | 6.1 | 23.17 |
| InstantID* | 7.12 | 7.63 | 10.65 | 6.23 | 24.51 |
| PuLID-SDXL | 7.43 | 8.02 | 10.96 | 7.28 | 26.26 |
| PuLID-FLUX* | 12.84 | 14.89 | 16.86 | 12.95 | 44.7 |
| InfineteYou* | 8.34 | 10.23 | 13.53 | 9.03 | 32.79 |

### A.4.9    User Study Template

Fig.24 show an example of our proposed user study on the performance of FreeCure.

Figure 24: **An example of user study.** We use scoring strategy to collect user assessments about the samples with/without FreeCure.

