# OpenReview forum: "Foundation Cures Personalization: Improving Personalized Models’ Prompt Consistency via Hidden Foundation Knowledge"
_NeurIPS.cc/2025/Conference — NeurIPS 2025 poster_

### Official Review · Reviewer_tTVJ · 2025-06-29

**Clarity:** 4
**Significance:** 3
**Originality:** 3
**Rating:** 4
**Confidence:** 5

**Summary:**

This paper aims to address the balance between identity preservation and prompt consistency in personalization diffusion models.
First, this paper figures out that the limited prompt consistency comes from the ID embeddings in cross-attention. To address it, this paper propose a training-free framework that first generate faces with/without id embeddings, and then modify the self-attention module to restore facial attributes. The proposed method can be seamlessly integrated into any personalization models, and bring a significant improvement on prompt alignment.

**Questions:**

Please address the weaknesses above, especially weaknesses 1 and 2.

**Ethical Concerns:**

["NO or VERY MINOR ethics concerns only"]

**Final Justification:**

Thanks for your detailed response. The rebuttal solves most of my concerns.

Regarding the cfg comparison, this paper applies the 2-way cfg for baselines (in Table 3). However, when we have multiple conditions, a common way is to use the 3-way cfg, which also has a branch that sets only one condition to empty (See Equation 3 and Appendix B in [A]). By adjusting the two cfg parameters, the prompt consistency and reference image preservation can be easily adjusted. I'd like to see the discussion between the proposed method and the 3-way cfg, not just the 2-way cfg baseline.


[A] Brooks, Tim, Aleksander Holynski, and Alexei A. Efros. "Instructpix2pix: Learning to follow image editing instructions." Proceedings of the IEEE/CVF conference on computer vision and pattern recognition. 2023.

**Limitations:**

yes

**Quality:**

3

**Strengths And Weaknesses:**

**Strengths**

1. The paper presents an insightful finding that well-tuned personalization models can retain the intrinsic capabilities of the foundation model. This suggests that degraded prompt consistency may stem from other sources rather than from the model itself — an observation that could influence future personalization strategies.

2. The proposed method  can be seamlessly integrated into any encoder-based tuning-free personalized models, suggesting strong generalization ability.

3. The results are promising, as it significantly improves the prompt consistency. Although the IF metric becomes lower, I totally understand, as this metric is biased by copy-paste behavior to some degree.

4. This paper provides clear technical details in terms of implementation and detailed ablation studies to verify the effectiveness each component.

**Weaknesses**

1. To my best knowledge, the balance between ID preservation and prompt consistency is a general  a general challenge beyond facial attributes. Thus, it is preferred that the proposed method can also. Currently, it is limited by the face parsing model and thus can only handle facial attributes. What happens for failure cases

2. The idea that using zero id embedding vectors is similar to the classifier-free guidance (CFG) strategy. Most personalization models already takes a 3-way strategy that set id vectors to 0 and text embeddings to zero separately. Thus, it would be better to provide details of the cfg setting in all proposed modules. For example, does the dual inference module already use 3-way cfg? Plus, this paper should also highlight the difference between cfg strategy and the proposed dual inference as they all share the idea of zero face embeddings.

3. It seems that the proposed method requires the identity embedding is injected into the model via cross-attention. It would be better if this paper could share some thoughts on full self-attention DiT models.

---

> ### Author Rebuttal · Authors · 2025-07-31
>
> Thanks for your constructive suggestions and positive comments on our insightful finding, strong generalization ability and promising results. We will raise the following points to address all your concerns.
>
> ## 1. More Diverse Segmentation Models and Attributes
> Thank you for raising this point. First, our method can leverage any attribute segmentation model to handle different types of features. In **Section A.2.4** of the supplementary materials, we discuss the performance of general semantic segmentation models (e.g., SAM) within our framework and address attributes that fall outside facial features. It is important to note that since most current encoder-based facial personalization methods extract and use the facial region as a conditional input, the copy-paste issue is most pronounced in facial attributes—which is the primary focus of our paper. In contrast, non-facial attributes, which are not subject to additional constraints during training, exhibit a less significant copy-paste effect.
>
> However, we fully understand your concern regarding the completeness of this discussion and have conducted supplementary experiments to address it. Following PuLID’s prompt set, we evaluated prompts involving common objects, clothing, and other typical non-facial attributes, using Segment Anything for mask generation. The experimental results across several baselines are as follows:
>
> > **Table 1: Additional Prompt Set with Non-facial Attributes**
> |   | new prompts                        |
> |---|------------------------------------|
> | 1 | a person wearing a *black headphone* |
> | 2 | a person wearing a *doctoral hat*    |
> | 3 | a person holding a *beer cup*        |
> | 4 | a person wearing a *spacesuit*       |
> | 5 | a person wearing a *yellow scarf*    |
>
> > **Table 2: Perfomance on Additional Prompt Set with Non-facial Attributes**
> |                           | PC    | IF    | Face Div. | IF x PC (hMean) |
> |---------------------------|-------|-------|-----------|-----------------|
> | Face2Diffusion            | 22.87 | 41.08 | 41.34     | 29.38           |
> | Face2Diffusion + FreeCure | **23.34 (+2.06%)** | 40.75 (-0.80%)| **41.91 (+1.38%)**    | **29.68(+1.01%)**|
> | InstantID                 | 22.43 | 65.55 | 47.73     | 33.42           |
> | InstantID + FreeCure      | **23.51(+4.81%)** | 65.34(-0.32%) | **48.06(+0.69%)**  | **34.58(+3.46%)**|
> | PhotoMaker                | 24.67 | 52.14 | 46.02     | 33.49           |
> | PhotoMaker + FreeCure     | **25.37(+2.84%)** | 51.77(-0.71%) | **46.58(+1.21%)**| **34.05(+1.67%)** |
> | PuLID (SDXL)              | 27.43 | 59.41 | 41.97     | 37.53           |
> | PuLID (SDXL) + FreeCure   | **28.13(+2.55%)** | 59.03(-0.64%) | **42.05(+0.19%)**| **38.10(+1.52%)**|
>
> Experimental results indicate that the facial personalization model exhibits a weaker copy-paste effect in non-facial regions, leading to less pronounced improvements in prompt consistency compared to the facial attributes specifically addressed in our submission. However, we observed that FreeCure still demonstrates significant restoration effects for attributes such as headphones and hats, which are in close proximity to the face (additional comparative images will be included in the updated version). Moreover, since the restoration occurs outside the facial region, FreeCure has a more negligible negative impact on identity fidelity. In summary, while our method remains applicable to various non-facial attributes due to its generalizability, the optimization effects are less pronounced but also noticeable compared to facial features.
>
> **Analysis of Failure Cases**. Although FreeCure shows consistent improvements in wide range of baselines and attributes. We observe that when enhancing transparent objects (e.g., cups), the restored images may exhibit optical distortions in transparent regions. We will include a discussion of this failure case in the Limitations section and explore potential optimizations for such cases in future work.
>
> ## 2. Comparison with Classifier-free Guidance
> We agree that this detail should be clarified. First, all our experiments do not modify the classifier-free guidance configuration in each baseline's official implementation, matching with our method's plug-and-play manner. That is to say, PD and FD process works when classifier-free guidance is included. For original inference with classifier-free guidance, the batch-wise information can be shown in the following table:
>
> > **Table 3: CFG Condition before Applying Dual Inference**
> | Batch Index| 0| 1|
> |--|--|--|
> | Condition Type | {${\emptyset}\_{id}, {\emptyset}\_{txt}$} | {${C}\_{id}, {C}\_{txt}$} |
>
> >  **Table 4: CFG Condition after Applying Dual Inference**
> | Batch Index    | 0| 1| 2| 3|
> |---|--|--|-----|--|
> | Condition Type | {${\emptyset}\_{id}, {\emptyset}\_{txt}$}  | {${\emptyset}\_{id}, {\emptyset}\_{txt}$}  | {${C}\_{id}, {C}\_{txt}$} | {${\emptyset}\_{id}, {C}\_{txt}$} |
>
> where ${C}\_{id}$ \& ${C}\_{txt}$ denote original identity and prompt conditions and ${\emptyset}\_{id}$ \& ${\emptyset}\_{txt}$ denote the null or zero-value conditions. This is similar to the situation that dual inference in running an inference with a batch size equal to 2 (the function of *chunk*() will operate correctly). Without the techniques, attention manipulation techniques like FreeCure, FD, and PD are independent. What should be emphasized is that such independence helps us to identify the foundational knowledge where high prompt consistency still exists in these personalization models.
>
> The core difference between CFG and FreeCure's dual inference lies in how they use zero face embeddings in their corresponding paradigm. First, during the training process of encoder-based personalization models with CFG, their basic target is still to reconstruct the reference faces. That is to say, even when the model receives a zero face embedding during some training step, their learnable modules (e.g., adapters, learnable embeddings) do not explicitly receive knowledge or constraints from the zero face embedding's denoising process. This also explains why nearly all encoder-based personalization models still face the copy-paste problem, even though the CFG strategy is introduced. In contrast, FreeCure uses zero face embedding to explicitly trigger information from the denosing process with zero face embedding (e.g., via FASA's self-attention map manipulation) and proposes a mechanism to inject such information into the original personalization process to improve its prompt consistency. We hope the explanation above can highlight the core difference and address your concerns.
>
> ## 3. Implementation details on Full-attention of DiTs
> Thanks for pointing out this. The implementation of FASA on DiT's full-attention is almost identical to U-Net's self-attention but with some slight differences in mask strategy. The input of full-attention is a joint sequence of noisy latent and text condition: $[X; C], X\in \mathbf{R}^{b\times l\_1 \times h}, C \in \mathbf{R}^{b \times l\_2 \times h}$. Since PD and FD share the identity text condition, we extract noisy latent relevant $K\_f^X, V\_f^X$ from the FD process and concatenate it to $K\_p^X, V\_p^X$ of the PD process, while $Q$ keeps consistent:
> $$
> \hat{Q\_p} = [Q\_p^X; Q\_p^C]; \hat{K\_p} = [K\_p^X;K\_p^C;K\_f^X]; \hat{V\_p} = [V\_p^X;V\_p^C;V\_f^X]
> $$
> The main difference lies in the mask strategy. Aligning with U-Net's implementation, we apply the all-one matrix to PD's original attention part. Specially, we assign scaling masks $\omega \mathcal{M}$ to attention map of $Q\_p^X and K\_f^X$ and assign a all-zero mask to attention map of $Q\_p^C and K\_f^X$ to preserve PD's original full-attention patterns:
> $$
> \mbox{FASA}(\hat{Q\_p}, \hat{K\_p}, \hat{V\_p}) = \mbox{Softmax}(\frac{\mathcal{M}(\omega)\_{flux}\odot \hat{Q\_{p}}\hat{K\_p}^{T}}{\sqrt{d}})\hat{V\_p}
> $$
>
> $$
> \mathcal{M}(\omega)\_{flux} = \begin{pmatrix}
>   \mathbf{1}\_{l\_1\times l\_1} & \mathbf{1}\_{l\_1\times l\_2} & \omega \mathcal{M}\_{l\_1\times l\_1} \\\\
>   \mathbf{1}\_{l\_2\times l\_1} & \mathbf{1}\_{l\_2\times l\_2} & \mathbf{0}\_{l\_2\times l\_1}
> \end{pmatrix}
> $$
> This mask pattern highly resembles FASA's implementation in U-Net, which injects attribute-aware features from FD to PD while keeping the original full-attention's pattern. We will update this formula for FLUX in the updated version of the submission.

---

### Official Review · Reviewer_kayu · 2025-07-02

**Clarity:** 3
**Significance:** 2
**Originality:** 2
**Rating:** 4
**Confidence:** 4

**Summary:**

This paper tries to tackle the common failure of personalization models to adhere to text prompts. It proposes FreeCure, a training-free framework that identifies a key cause: identity embeddings suppressing the foundation model's prompt-following ability. FreeCure uses a dual-inference process to leverage the "cured" output from the foundation model to guide and correct the personalized generation, improving prompt consistency while maintaining identity.

**Questions:**

1. How much does the cost of inference increase?
2. How robust are the attributes parsing and segmentation models? Can complex user prompts be automatically processed?
3. Figure  3 shows the implementation of Unet, how is the text-image full attention model like flux implemented?

**Ethical Concerns:**

["NO or VERY MINOR ethics concerns only"]

**Final Justification:**

Thanks for the rebuttal; it effectively solved my question， and I raise my score to weak accept

**Limitations:**

yes

**Quality:**

2

**Strengths And Weaknesses:**

1. The argument that ID embeddings suppress the model's underlying prompt-following capabilities is reasonable and provides a strong motivation for the solution.
2. The training-free "curing" approach is elegant and shows significant, consistent improvements across a wide range of popular personalization models and diverse foundation models.

Weaknesses:
1. Increased Inference Cost: The dual-inference design effectively doubles the computational cost per step, which is a significant trade-off for speed-sensitive applications.
2. Relies on an external segmentation model for localizing attributes, which adds a layer of complexity and a potential point of failure.
3. The two proposed points appear to be more of practical engineering techniques.

---

> ### Author Rebuttal · Authors · 2025-07-31
>
> ## 1. Inference Cost
>
> We conduct experiments on a single H800 GPU with 80 GB VRAM to measure each baseline’s inference time before and after applying FreeCure. The table below reports the results ("*" means 30-step denoising due to official guidance and 50-step denoising is applied for the rest baselines):
>
> >**Table 1: Runtime Analysis**
> |     | Baseline (seconds) | Baseline + FreeCure (Seconds) | |   |  |
> |:----:|:--------:|:---------:|:------:|:-------:|:--------:|
> |     |  **Total**| Stage 1| Stage 2 FASA | APG   | **Total**  |
> | FastComposer |**1.79**| 1.96| 2.19| 1.83  | **5.98** |
> | Face-Diffuser   |**2.13**| 2.63| 2.33 | 1.92  | **6.88**|
> | Face2Diffusion* |**1.13**| 1.25| 1.47| 1.23  | **3.95** |
> | PhotoMaker     |**6.52**| 7.04| 10.03| 6.1   | **23.17** |
> | InstantID*          |**7.12**| 7.63| 10.65| 6.23  |**24.51** |
> | PuLID-SDXL    |**7.43**| 8.02| 10.96| 7.28  | **26.26**  |
> | PuLID-FLUX*    |**12.84**| 14.89| 16.86| 12.95 | **44.7**  |
> | InfineteYou*      | **8.34**| 10.23| 13.53| 9.03  | **32.79**  |
>
> While we acknowledge that introducing extra calculation can lead to longer runtime, we also emphasize that most training-free methods introduce extra processes, including inversion operation, denoising processes, and extra attention calculation. Plus, comparing to the fact that most encoder-based personalization methods require a long range time of data collection, preprocessing and tuning (this may consume a large array of GPUs and thousands of GPU-hours), our training-free method provides an innovative perspective for prompt consistency improvement by the knowledge in themselves, without the need for designing new model architecture and dataset curation.
>
> ## 2. Robustness of Segmentation Model
> Thanks for pointing this out. We would like to address your concern from two aspects:
>
> + **Robustness of FreeCure when dealing with inaccurate masks.** FreeCure demonstrate consistent performance even when masks are inaccurate. To validate this, we design two-way validation experiments that introduce inaccuracy to attribute masks. First, we apply a dilation operation on masks, which might include some area which does not belong to the target attribute area. Second, we apply an erosion operation, which might cause loss of some useful information about attributes. To create significant distortion for masks, we set the receptive field of both dilation and erosion to 25 pixels. We conduct experiments on several baselines, and the table below reports the results for respective prompt consistency (PC) and ideneity fidelity (IF):
>
> > **Table 2: PC and IF Performance for Inaccurate Masks**
> |            | PC with original masks | IF with original masks | PC with dilated masks | IF with dilated masks | PC with eroded masks | IF with eroded masks |
> |---|----|------|---|----|----|----|
> | InstantID  | 23.62    | 62.01   | 23.45  | 62.23  | 23.21  | 62.23  |
> | PhotoMaker | 24.91   | 50.15   | 24.87  | 50.01    | 24.34 | 50.44   |
> | PuLID-SDXL | 26.05  | 56.95  | 25.51  | 56.34 | 25.36  | 57.01 |
> | PuLID-FLUX | 24.78   | 72.61  | 24.52   | 72.39   | 24.32  | 72.97  |
>
> The results demonstrate that identity fidelity (IF) remains largely consistent across both dilated and eroded mask conditions, with slight improvements observed in eroded masks. This occurs because mask erosion preserves certain attributes similar to the reference image. As for prompt consistency, both dilation and erosion cause minimal degradation, though erosion exhibits marginally greater impact. We consider that this stems from cases where fine-grained attribute restoration (e.g., iris color, earrings) may fail when the mask completely degenerates due to erosion.
>
> Generally, the performance degradation of FreeCure due to inaccurate masks is limited. Moreover, in practical applications, specialized segmentation models (e.g., BiSeNet, Segment Anything) exhibit high robustness, making failure cases caused by mask inaccuracies even rarer than observed in these deliberately designed experiments. This further validates the inherent robustness of the FreeCure framework.
>
> + **Automatic Extraction for Attribute Masks.** FreeCure’s mask extraction process is auto-driven without any case-specific design.  Our system employs a lookup table encompassing a broad spectrum of facial attributes or other attributes to align with the input prompt. When target attributes are matched, the face parsing model (e.g., BiSeNet) or Segment Anything Model (SAM) automatically generates the corresponding masks in a fully automated manner. Specifically, BiSeNet inherently predicts masks for all facial attributes, so only the matched masks need to be extracted. In the Segment Anything configuration, the model takes the matched attributes as input to produce the final detection mask. A merging process will gather all masks together and generate a mask relevant to all attributes:
> $$
> \mathcal{M}(j,k) = max(\mathcal{M}_{i}(j,k)).
> $$
> Thanks to these models’ expertise and generalization capabilities in segmentation, the entire process operates without manual intervention, eliminating the need for specialized handling of particular attributes.
>
> ## 3. Engineering Implementation
> First, as discussed in **Section 3** of our submission, our architectural innovation on self-attention is grounded in solid theoretical foundations: the application of attention theory in diffusion models – where cross-attention controls identity injection while self-attention regulates spatial feature arrangements. This constitutes FreeCure’s core rationale for achieving attribute improvement without compromising identity fidelity. Thus, our method is driven by clear motivations and rigorous theoretical reasoning, rather than accidental parameter tuning or architectural trial-and-error. Second, our approach does not rely on ad hoc parameter design, demonstrating its generalizability as a performance-enhancing framework rather than case-specific engineering. As evidenced throughout our experiments, we maintained identical hyperparameter configurations across all baselines ($\omega=2.0$ and $\gamma=0.5$). These two aspects collectively distinguish our method from engineering-driven solutions.
>
> ## 4. Implementation details on FLUX
> Thanks for pointing out this. The implementation of FASA on FLUX's full-attention is almost identical to U-Net's self-attention but with some slight differences in mask strategy. The input of full-attention is a joint sequence of noisy latent and text condition: $[X; C], X\in \mathbf{R}^{b\times l\_1 \times h}, C \in \mathbf{R}^{b \times l\_2 \times h}$. Since PD and FD share the identity text condition, we extract noisy latent relevant $K\_f^X, V\_f^X$ from the FD process and concatenate it to $K\_p^X, V\_p^X$ of the PD process, while $Q$ keeps consistent:
> $$
> \hat{Q\_p} = [Q\_p^X; Q\_p^C]; \hat{K\_p} = [K\_p^X;K\_p^C;K\_f^X]; \hat{V\_p} = [V\_p^X;V\_p^C;V\_f^X]
> $$
> The main difference lies in the mask strategy. Aligning with U-Net's implementation, we apply the all-one matrix to PD's original attention part. Specially, we assign scaling masks $\omega \mathcal{M}$ to attention map of $Q\_p^X and K\_f^X$ and assign a all-zero mask to attention map of $Q\_p^C and K\_f^X$ to preserve PD's original full-attention patterns:
> $$
> \mbox{FASA}(\hat{Q\_p}, \hat{K\_p}, \hat{V\_p}) = \mbox{Softmax}(\frac{\mathcal{M}(\omega)\_{flux}\odot \hat{Q\_{p}}\hat{K\_p}^{T}}{\sqrt{d}})\hat{V\_p}
> $$
>
> $$
> \mathcal{M}(\omega)\_{flux} = \begin{pmatrix}
>   \mathbf{1}\_{l\_1\times l\_1} & \mathbf{1}\_{l\_1\times l\_2} & \omega \mathcal{M}\_{l\_1\times l\_1} \\\\
>   \mathbf{1}\_{l\_2\times l\_1} & \mathbf{1}\_{l\_2\times l\_2} & \mathbf{0}\_{l\_2\times l\_1}
> \end{pmatrix}
> $$
> This mask pattern highly resembles FASA's implementation in U-Net, which injects attribute-aware features from FD to PD while keeping the original full-attention's pattern. We will update this formula for FLUX in the updated version of the submission.

---

> > ### Comment · Reviewer_kayu · 2025-08-02
> >
> > Thanks for the rebuttal; it effectively solved my question， and I raise my score to weak accept

---

> > > ### Author Response · Authors · 2025-08-02
> > >
> > > Thanks for your valuable time and constructive feedback. We highly appreciate your support.
> > >
> > > Best wishes,
> > >
> > > Authors of Submission 15942

---

### Official Review · Reviewer_nf3e · 2025-07-03

**Clarity:** 3
**Significance:** 3
**Originality:** 2
**Rating:** 4
**Confidence:** 5

**Summary:**

This paper introduces a training-free framework called FreeCure attempted to address the common trade-off between identity fidelity and prompt consistency in personalized image generation.
The core of their method, Foundation-Aware Self-Attention (FASA), transfers features from the foundation diffusion (FD) process to the personalization diffusion (PD) process using segmentation masks to improve attribute control while aiming to preserve the subject's identity. Additionally, an Asymmetric Prompt Guidance (APG) technique is used to refine abstract attributes like expressions. The method is presented as a plug-and-play module for various existing personalization models.

**Questions:**

1. Novelty of the FASA Technique: The writing in the methodology section is misleading. The proposed Foundation-Aware Self-Attention (FASA) mechanism appears to be a rebranding of the "mutual self-attention" concept introduced in the MASACtrl paper. The core operation of concatenating key and value matrices from two different generation processes to control synthesis is not a novel contribution.

2. Missing Mathematical Equation of APG: I am not sure if I understand this correctly. As I interpret it, you first run FD alone in the first pass to obtain intermediate denoised latents without the expression prompt, and then use these latents as the starting point for a second pass with FASA and the expression prompt. Is this correct? Why is inversion necessary? Why not run the second step directly? Is the choice of $\gamma$ consistent across all your results? Do you use APG in all results shown in the paper?

3. Attribute Entanglement: The shared attention mechanism is fundamentally based on KV injection. The prior identity information from the hidden states of the FD process will also be introduced into the PD process, leading to lower identity preservation, as observed in nearly all samples.

4. Mask Dependency: The masking strategy must be tuned for each prompt and relies on detailed manual masking. How to eliminate manual masking while keeping the system stable should be further investigated.

5. Missing Discussion of Critical Prior Art: The paper fails to discuss or compare against "Mixture-of-Attention for Subject-Context Disentanglement in Personalized Image Generation" (MoA), presented at SIGGRAPH Asia 2024. MoA can be considered a trainable counterpart addressing a similar problem as this work.

6. Missing Baseline Comparison: A simple yet effective baseline would be to compare FreeCure against a state-of-the-art personalization method using a reduced inference weight (e.g., a scale of 0.6) combined with a post-processing face-swapping technique. This alternative also aims to balance identity preservation and prompt fidelity and would serve as a valuable comparison point given the complexity introduced by FreeCure.

**Ethical Concerns:**

["NO or VERY MINOR ethics concerns only"]

**Final Justification:**

At current, I would like to raise my score to a borderline accept. I appreciate the efforts of authors in clarifying some of my previous concerns. My only remaining concern is the Attribute Entanglement in Prompt Decomposition (PD). While the rebuttal argues that masking alleviates entanglement, my concern is that the key-value pairs from PD still contain features entangled with both target attributes and unwanted prior identity signals. These prior identity features are not explicitly excluded during shared attention. I believe that further disentangling and removing the prior identity features before injection would strengthen the proposed method.

**Limitations:**

The limitation discussion of this work requires more clarification. The authors state that FreeCure’s performance is influenced by the characteristics of the underlying personalization models, including their inherent biases and maximum achievable capabilities. This explanation is reasonable and expected. However, we would appreciate a more candid discussion of the general limitations of the proposed method — even when applied within a fixed foundation model. I provide some limitations are raised in the questions above and the authors might know more limitations and can showcase more failure cases.

**Quality:**

3

**Strengths And Weaknesses:**

Strengths:
The main strength of this work lies in its introduction of a training-free approach that addresses a known limitation of personalization methods — namely, the performance drop in prompt alignment.

Weaknesses:
However, in nearly all examples, the proposed method suffers from a clear trade-off between identity preservation and prompt alignment. Additionally, the shared attention mechanism used is not novel; similar techniques have been widely explored in prior work within the field. See Questions section for more details.

---

> ### Author Rebuttal · Authors · 2025-07-30
>
> ## 1. Novelty Clarification
> While we agree that both FreeCure and MasaCtrl apply self-attention manipulation. We argue that our method is significantly different from MasaCtrl in several points that guarantee its novelty:
> + **Different Generation Paradigm.** Masactrl converts an input image into a latent code through inversion and then copies it to its dual inference counterpart. In contrast, FreeCure exhibits less dependency on this. FreeCure's primary process generates images directly from noise, and the initial noise for its FD and PD processes can be sampled arbitrarily without requiring consistency. This fundamental difference in the generative paradigm enhances the diversity of restoration results in FreeCure (see **Section 5.2 and A.3.2** of the submission for details). By comparison, MasaCtrl's dual inference mechanism strictly depends on identical latent codes, lacking exploration into the effects of different initial latents.
> + **Different Task Focus.** While MasaCtrl primarily investigates non-rigid image editing tasks such as changing pose or viewpoint, our work focuses on enhancing prompt consistency in facial personalization models. Moreover, whereas MasaCtrl performs editing on the entire subject (e.g., pose adjustment), our focus lies in refining internal features (regardless of their granularity) of the subject and can seamlessly integrate with various mainstream facial personalization models.
> + **Better Adaptability.** The adaptability of the FreeCure extends beyond U-Net's self-attention mechanism. Its FASA algorithm can also be applied to the full-attention of FLUX. However, this aspect remains unexplored in MasaCtrl’s discussions. MasaCtrl also lacks discussion on the simultaneous editing of multiple object attributes. In contrast, our proposed FASA framework can effectively handle multiple facial attributes concurrently while maintaining high identity fidelity.
> + **Masking Strategy.** Building upon the second point, while MasaCtrl’s mutual self-attention mechanism leverages cross-attention maps as guidance, its subsequent thresholding operation introduces great noise and fails to accurately localize fine-grained features. To further validate this, we modify FreeCure's mask by replacing segmentation-model-extracted masks with those derived from cross-attention maps. We aggregated cross-attention maps for the corresponding tokens in a way identical to MasaCtrl and applied Otsu’s thresholding method for binarization:
>
> > **Table 1: Performance with Cross-attention Mask**
> || PC of Original FreeCure|IF of Original FreeCure|PC of FreeCure with cross-attention mask|IF of FreeCure with cross-attention mask|
> |-|-|-|-|-|
> |InstantID|**23.62**|**62.01**|19.23|59.23|
> |PhotoMaker|**24.91**|**50.15**|22.43|49.32|
> |PuLID-SDXL|**26.05**|**56.95**|24.52|53.43|
> |PuLID-FLUX|**24.78**|**72.61**|22.95|67.67|
>
> The generated images will be updated in the new version. We can see that when using the cross-attention map as a mask, all baselines showed decreased performance in both prompt consistency and identity fidelity. This happens because compared to masks created by expert segmentation models, the cross-attention mask picks up too much irrelevant noise during the enhancement process. This noise undermines useful features and damages the original facial identity fidelity.
>
> To sum up, we argue that although FASA of FreeCure and mutual self-attention of MasaCtrl share a similar concept of attention control, their difference lies in, but is not limited to, overall paradigm, problem focus, adaptability, as well as application complexity. Since other reviewers (uVNV,rzYz, and tTVJ) acknowledge the motivation, simplicity, and generalizability of FreeCure, its novelty and contribution should be guaranteed.
>
> ## 2. Supplementary Explanation of APG
> + **Formula and implementation of APG.** During APG, FASA is excluded since this stage only involves single-batch inference of the baseline model. We employ the original baseline's FD process independently and leverage its inherent consistency in handling abstract attributes to further refine the results improved by FASA. The process can be described via the following formula:
> $$
> \hat{z_{\gamma T}} = \mbox{Inversion}(I_r, c_0);
> I_{out} = \mbox{Denoise}(\hat{z_{\gamma T}}, c_1)
> $$
> where $I\_r, I\_{out}, \hat{z\_{\gamma T}}$ matches the annotation in Figure 3 of submission, $c\_0$ denotes a template prompt without any attribute description (e.g., "*a man*") and $c\_1$ denotes prompt containing target attirbutes (e.g., "*a man laughing*").
>
> + **Reason for separating APG from the second step.** Such separation allows distinct handling of spatially localized features and abstract attributes. Processing these features jointly within the same denoising stage would hinder their effective decoupling, ultimately compromising the enhancement of individual characteristics.
>
> + **Value selection for $\gamma$.** All experimental data and visual results in our main submission were obtained consistently using $\gamma=0.5$. While we acknowledge that the optimal $\gamma$ may vary slightly across different baselines, its value consistently falls within the range of $[0.5, 0.6]$. For a plug-and-play method, we argue that this minor variation is acceptable due to the diversity of baselines.
>
> + **Usage of APG**. During the generation phase, if no prompt related to abstract features (e.g., facial expression) is matched, the APG module can be safely omitted.
>
> ## 3. Attribute Entanglement
> We argue that the comments of "*trade-off in nearly all examples*" do not have specific evidence. The problem of "*...The prior identity information from the hidden states of the FD process will also be introduced...*" can be directly answered by **Figure 4** of our submission. Our mask guidance mechanism provides an effective solution to protect the identity fidelity of PD from FD. Our visual experiments in **Section 5.2** also confirm that in non-target regions, the FD process exerts minimal influence on the PD denoising process. Plus, our method's superiority has been validated both qualitatively and quantitatively and acknowledged by other reviewers. A supplementary user study required by Reviewer uVNV also highly aligns with our consistent improvement of the submission. Therefore, we respectfully suggest that you reconsider your assessment.
>
> ## 4. Mask Dependency
> You may have some misunderstandings which we would like to clarify: FreeCure’s mask extraction does not rely on any case-specific "*tuning*" or "*manual processing*". Instead, our system employs a lookup table encompassing various facial attributes to align with the input prompt. When target attributes are matched, the face parsing model (e.g., BiSeNet) or Segment Anything Model (SAM) automatically generates the corresponding masks. Specifically, BiSeNet inherently predicts masks for all facial attributes, so only the matched masks need to be extracted. In the Segment Anything, the model takes the matched attributes as input to produce the mask. Thanks to these models’ generalization capabilities in segmentation, the entire process operates without manual intervention.
>
> ## 5. Discussion of MoA
> MoA's prior attention primarily addresses the generation of regions unrelated to the subject (e.g., background). This objective fundamentally differs from FreeCure, which is designed to enhance prompt consistency for features within the main subject. The only commonality between the two approaches may be their shared goal of integrating foundational model knowledge into personalized models. Additionally, FreeCure is a plug-and-play framework, compatible with diverse personalization methods. Its relationship with MoA is better characterized as complementary (i.e., upstream/downstream) rather than being a "*trainable counterpart*". Given FreeCure’s demonstrated adaptability across a wide range of personalization models, we anticipate its potential application to MoA as well. In the updated version of our submission, we will include this reference and discuss the feasibility of integrating FreeCure into MoA.
>
> ## 6. Missing Baseline Comparison
> We argue that the pipeline proposed by the reviewer cannot be comparable to our method. The reviewer suggests using a '*reduced inference weight*' to generate results, but this approach harms both identity fidelity and prompt consistency (see our discussion in **Section 3**). Instead, directly adopting our proposed FD process (equivalent to setting the 'inference weight' directly to 0) could be a more viable solution.
>
> Plus, this pipeline critically depends on face swapping models to disentangle identity and attributes, particularly when dealing with occlusions (e.g., sunglasses, masks) or spatial misalignments in foundation model outputs. Current art face-swapping methods (e.g., DreamID, DiffSwap) rarely address these cases, and their failures mirror the copy-paste issues in personalized models. In contrast, FreeCure provides a simpler, more effective solution, as detailed in **Section 5.2**.
>
> In summary, there exists a significant gap between the baseline mentioned by the reviewer and the problem setting addressed in our work. We contend that, in the context of facial personalization, this pipeline does not constitute a comparable paradigm to our proposed method.
>
> ## 7. Limitations
> Although FreeCure shows consistent improvements in wide range of baselines and attributes. We observe that when enhancing transparent objects (e.g., cups), the restored images may exhibit optical distortions in transparent regions. We will include a discussion of this failure case in the Limitations section and explore potential optimizations for such cases in future work.
>
> ## 8. Ethical Concerns
> Since our evaluation dataset comes from public resources (e.g., CelebA-HQ, public webpages). It is safe to say that concerns on data privacy is minimal.
>
> ## Reference
> [1] MoA, SIGGRAPH Asia 2024.
>
> [2] DreamID, arXiv 2025.
>
> [3] Diffswap, CVPR 2023.

---

> > ### Comment · Reviewer_nf3e · 2025-08-03
> > **Reply to the rebuttal**
> >
> > Hi authors,
> >
> > Thank you for your rebuttal. After reading it and the other reviews, I would like to raise my score to a borderline accept. I appreciate your efforts in clarifying several points and would like to offer the following suggestions and concerns for your consideration:
> >
> > Suggestion – Related Work on Attention Manipulation. I recommend expanding the related work section to include more recent advances in attention manipulation. Notably, works such as MasaCtrl, MoA, and Nested Attention are relevant to your methodology and would provide valuable context for readers.
> >
> > Concern – Attribute Entanglement in Prompt Decomposition (PD). While the rebuttal argues that masking alleviates entanglement, my concern is that the key-value pairs from PD still contain features entangled with both target attributes and unwanted prior identity signals. These prior identity features are not explicitly excluded during shared attention. I believe that further disentangling and removing the prior identity features before injection would strengthen the proposed method. Any thoughts? or disagreement?

---

> > > ### Author Response · Authors · 2025-08-03
> > >
> > > Dear Reviewer nf3e,
> > >
> > > Thank you for your valuable time and positive assessment of our work following the rebuttal. We sincerely appreciate your feedback and will incorporate your suggestions into our updated submission. Below are the key improvements we will make:
> > >
> > > + **Related Work on Attention Manipulation**. We will expand our literature review to include discussion of relevant works (e.g., MasaCtrl, MoA, and Nested Attention) to provide readers with a more comprehensive understanding of attention manipulation techniques.
> > >
> > > + **Attribute Entanglement in Prompt Decomposition**. As you noted, identity-relevant information may still be entangled in the masked attention regions in some certain cases. We will add the discussion of this in the Limitations & Future Work section.
> > >
> > > Again, we appreciate your insightful suggestions, which have greatly helped improve our submission.
> > >
> > > **Expanded Literatures**
> > > > [1] MoA, SIGGRAPH Asia 2024\
> > > > [2] Nest Attention, SIGGRAPH 2025
> > >
> > > Best regards, \
> > > Authors of Submission 15942

---

### Official Review · Reviewer_rzYz · 2025-07-03

**Clarity:** 4
**Significance:** 3
**Originality:** 4
**Rating:** 5
**Confidence:** 3

**Summary:**

The authors introduce the tradeoff between identity preservation and prompt adherence in current human-face embedding-based personalization methods, provide an explanation and analyze the effect of key related factors - the aggressive effect of personalization embeddings on cross attention layers, and the relative sensitivity of these layers to any change. Then, they provide a method that avoids cross-attention manipulation by taking self attention layers into account. They present a method that harnesses the spatial and semantic information encapsulated in the non-personalized foundation models self attention layers to enforce the wanted changes in the personalized generation process, without harming the personalization itself.

**Questions:**

* The trade-off depicted in the right rows Figure 2 is clear, but a finer resolution is needed to contradict the existence of a well optimized sweet spot. For example, in the top-row between alpha=0.2 and alpha=0.4, and in the bottom-row between alpha=0.4 and alpha=0.6.
* In section 4.1 (FASA Mechanism), there is a change in the matrices’ dimensionality due to concatenation. How does this change incorporate in the existing architecture? It might be implicitly inferred with previous domain knowledge, but I believe it requires explicit addressing.
* The method details the manipulation it applies to the UNET inference process, but some of the presented results are derived from FLUX based methods, while FLUX does not contain UNET in its architecture. This part needs clarification.
* The set of desired prompts is rather small - curly hair (in different colors), eye color, happiness, sunglasses, and pearl earrings. I would like to see a wider variety of prompts with non-trivial specifications , e.g. with an eye path, posture change, background environment, etc.

**Ethical Concerns:**

["NO or VERY MINOR ethics concerns only"]

**Final Justification:**

Sorry for missing this field, please see my comments  in the author discussion

**Limitations:**

Yes

**Quality:**

4

**Strengths And Weaknesses:**

Strengths:
* The paper includes a convincing description of the problem, with a well performed empirical analysis to support it.
* The method is innovative and yet not too complicated.
* The results, both quantitatively and qualitatively,  show consistent improvement compared to baselines.
* The framework is applicable to multiple embedding-based methods, spanning different base models.
Weaknesses:
* The paper presents many different faces (mainly in the appendix), but the variety of prompts is low.
* The analysis in section 3 could be more complete.
* Some technical details are missing, or de-emphasized.

---

> ### Author Rebuttal · Authors · 2025-07-30
>
> Thanks for your valuable comments and positive evaluation on our empirical analysis, method simplicity and experimental results. We propose the following points to address all your concerns:
> ## 1. More Comprehensive Prompt Set
> Thank you for raising this point. First, our method can leverage any attribute segmentation model to handle different types of features. In **Section A.2.4** of the supplementary materials, we discuss the performance of general semantic segmentation models (e.g., SAM) within our framework and address attributes that fall outside facial features. It is important to note that since most current encoder-based facial personalization methods extract and use the facial region as a conditional input, the copy-paste issue is most pronounced in facial attributes—which is the primary focus of our paper. In contrast, non-facial attributes, which are not subject to additional constraints during training, exhibit a less significant copy-paste effect.
>
> However, we fully understand your concern regarding the completeness of this discussion and have conducted supplementary experiments to address it. Following PuLID’s prompt set, we evaluated prompts involving common objects, clothing, and other typical non-facial attributes, using Segment Anything for mask generation. The experimental results across several baselines are as follows:
>
> > **Table 1: Additional Prompt Set with Non-facial Attributes**
> |   | new prompts                        |
> |---|------------------------------------|
> | 1 | a person wearing a *black headphone* |
> | 2 | a person wearing a *doctoral hat*    |
> | 3 | a person holding a *beer cup*        |
> | 4 | a person wearing a *spacesuit*       |
> | 5 | a person wearing a *yellow scarf*    |
>
> > **Table 2: Perfomance on Additional Prompt Set with Non-facial Attributes**
> |                           | PC    | IF    | Face Div. | IF x PC (hMean) |
> |---------------------------|-------|-------|-----------|-----------------|
> | Face2Diffusion            | 22.87 | 41.08 | 41.34     | 29.38           |
> | Face2Diffusion + FreeCure | **23.34 (+2.06%)** | 40.75 (-0.80%)| **41.91 (+1.38%)**    | **29.68(+1.01%)**|
> | InstantID                 | 22.43 | 65.55 | 47.73     | 33.42           |
> | InstantID + FreeCure      | **23.51(+4.81%)** | 65.34(-0.32%) | **48.06(+0.69%)**  | **34.58(+3.46%)**|
> | PhotoMaker                | 24.67 | 52.14 | 46.02     | 33.49           |
> | PhotoMaker + FreeCure     | **25.37(+2.84%)** | 51.77(-0.71%) | **46.58(+1.21%)**| **34.05(+1.67%)** |
> | PuLID (SDXL)              | 27.43 | 59.41 | 41.97     | 37.53           |
> | PuLID (SDXL) + FreeCure   | **28.13(+2.55%)** | 59.03(-0.64%) | **42.05(+0.19%)**| **38.10(+1.52%)**|
>
> Experimental results indicate that the facial personalization model exhibits a weaker copy-paste effect in non-facial regions, leading to less pronounced improvements in prompt consistency compared to the facial attributes specifically addressed in our submission. However, we observed that FreeCure still demonstrates significant restoration effects for attributes such as headphones and hats, which are in close proximity to the face (additional comparative images will be included in the updated version). Moreover, since the restoration occurs outside the facial region, FreeCure has a more negligible negative impact on identity fidelity. Regarding attributes such as body pose and background, we believe their relevance to our paper’s focus—resolving face copy-paste issues—is minimal, and thus they were not discussed. We appreciate your understanding in this regard.
>
> In summary, while our method remains applicable to various non-facial attributes due to its generalizability, the optimization effects are less pronounced but also noticeable compared to facial features.
>
> ## 2. Detailed discussion for Section 3
> Thanks for pointing out this detail. We have conducted a detailed analysis using finer-grained $\alpha$ values and their corresponding quantitative outcomes. Below are the results:
>
> > **Table 3: Fined-grained Quantitative Metrics for Examples of Cross-attention Interpolation in Section 3**
> |            | $\alpha$              | 0     | 0.1   | 0.2   | 0.3   | 0.4   | 0.5   | 0.6   | 0.7   | 0.8   | 0.9   | 1     |
> |------------|--------------------|-------|-------|-------|-------|-------|-------|-------|-------|-------|-------|-------|
> | top row    | IF  | 50.23 | 37.43 | **24.53** | **16.42** | **10.34** | 7.32  | 6.42  | 4.22  | 3.43  | 2.34  | 2.1   |
> |            | PC | 24.98 | 25.39 | **25.5**  | **25.61** | **26.39** | 27.53 | 28.54 | 29.43 | 30.43 | 32.34 | 33.01 |
> | bottom row | IF  | 52.34 | 50.23 | 49.52 | 40.01 | **29.12** | **23.89** | **15.75** | 10.34 | 6.32  | 4.11  | 3.2   |
> |            | PC | 26.76 | 28.43 | 30.34 | 31.12 | **31.09** | **31.87** | **32.98** | 33.43 | 34.23 | 34.79 | 35.02 |
>
> The results demonstrate that identity fidelity declines sharply even when only a small portion of the original cross-attention map is altered, suggesting that identifying a '*well-optimized sweet spot*' through cross-attention manipulation alone is highly challenging. Plus, as you noted, the fine-grained performance varies considerably between different $\alpha$ ranges (e.g., $\alpha \in [0.2, 0.4]$ for the top row and $\alpha \in [0.4, 0.6]$), further highlighting a key limitation of cross-attention manipulation: its optimal control parameter can exhibit substantial variability. We will include a more detailed version of Figure 2 in the updated submission to provide further clarity.
>
> ## 3. Matrices Dimensionality Change of FASA
> We reflect this question by providing the details of FASA. In the original self-attention mechanism with batch size 1, the dimensionality of $Q, K, V$ considering with $n$ heads is:
> $$
> Q, K, V \in \mathbf{R}^{1\times n \times l \times (\frac{h}{n})}
> $$
> The calculation of attention map $A = Softmax(\frac{QK^T}{\sqrt{d}}) \in \mathbf{R}^{1\times n \times l \times l}$ is straightforward. After we replace this self-attention layer with FASA module, typically we have batch size 2 due to dual inference design:
> $$
>     \hat{Q}, \hat{K}, \hat{V} \in \mathbf{R}^{2\times n \times l \times (\frac{h}{n})}
> $$
> In FASA inference process, the PD process is in the first batch while FD is in the second. We copy the FD batch and concatenate it to PD batch along the sequence dimension $l$:
> $$
>     \hat{K}\_{[0, :, :, :]} \leftarrow \mbox{concat}(\hat{K}\_{[0, :, :, :]}, \hat{K}\_{[1, :, :, :]}, \mbox{dim=2})
> $$
> $$
>     \hat{V}\_{[0, :, :, :]} \leftarrow \mbox{concat}(\hat{V}\_{[0, :, :, :]}, \hat{V}\_{[1, :, :, :]}, \mbox{dim=2})
> $$
> Now the PD process have an extended matrices of $\hat{K}\_{p}, \hat{V}\_{p} \in \mathbf{R}^{1\times n \times 2l \times (\frac{h}{n})}$. Therefore, FASA's attention map matrices $\hat{A\_p} \in \mathbf{R}^{1\times n \times l \times 2l}$ should also be extended. As for the scaling mask strategy, we resize the extracted mask $\mathcal{M}$ to the size $l\times l$ and replicate it to target size $1\times n \times l\times l$ after concatenating with a matrix with all elements equal to 1; this new mask can process $\hat{A\_p}$ accordingly. The entire process of PD process after applying FASA is as follows:
> $$
>     \mbox{FASA}(\hat{Q\_p}, \hat{K\_p}, \hat{V\_p}) = \mbox{Softmax}(\frac{[\mathbf{1},\omega\mathcal{M}]\odot \hat{Q\_{p}}\hat{K\_p}^{T}}{\sqrt{d}})\hat{V\_p}.
> $$
> This process aligns with Eq. 4 in the submission. Afterwards, PD's output dimensionality aligns with the original FD process and can be gathered together for the inference process of subsequent layers (e.g., FFN). Please note that since the PD and FD's matrices of $K$ and $V$ have different dimensions, their attention scores are calculated separately. However, this is easy to implement because the communication happens between two batches, and the external mask $\mathcal{M}$ can be defined during FASA modules' initialization.
>
> ## 4. FASA's Implementation in FLUX
> Thanks for your comments on this. We agree that this point should be clarified. The implementation of FASA on FLUX's full-attention is almost identical to U-Net's self-attention but with some slight differences in mask strategy. The input of full-attention is a joint sequence of noisy latent and text condition: $[X; C], X\in \mathbf{R}^{b\times l\_1 \times h}, C \in \mathbf{R}^{b \times l\_2 \times h}$. Since PD and FD share the identity text condition, we extract noisy latent relevant $K\_f^X, V\_f^X$ from the FD process and concatenate it to $K\_p^X, V\_p^X$ of the PD process, while $Q$ keeps consistent:
> $$
> \hat{Q\_p} = [Q\_p^X; Q\_p^C]; \hat{K\_p} = [K\_p^X;K\_p^C;K\_f^X]; \hat{V\_p} = [V\_p^X;V\_p^C;V\_f^X]
> $$
> The main difference lies in the mask strategy. Aligning with U-Net's implementation, we apply the all-one matrix to PD's original attention part. Specially, we assign scaling masks $\omega \mathcal{M}$ to attention map of $Q\_p^X and K\_f^X$ and assign a all-zero mask to attention map of $Q\_p^C and K\_f^X$ to preserve PD's original full-attention patterns:
> $$
> \mbox{FASA}(\hat{Q\_p}, \hat{K\_p}, \hat{V\_p}) = \mbox{Softmax}(\frac{\mathcal{M}(\omega)\_{flux}\odot \hat{Q\_{p}}\hat{K\_p}^{T}}{\sqrt{d}})\hat{V\_p},
> $$
> $$
> \mathcal{M}(\omega)\_{flux} = \begin{pmatrix}
>   \mathbf{1}\_{l\_1\times l\_1} & \mathbf{1}\_{l\_1\times l\_2} & \omega \mathcal{M}\_{l\_1\times l\_1} \\\\
>   \mathbf{1}\_{l\_2\times l\_1} & \mathbf{1}\_{l\_2\times l\_2} & \mathbf{0}\_{l\_2\times l\_1}
> \end{pmatrix}
> $$
> This mask pattern resembles FASA's implementation in U-Net which injects attribute-aware features from FD to PD while keeping the original full-attention's pattern. We will update this formula for FLUX in the updated version of the submission.

---

> > ### Author Response · Authors · 2025-08-05
> > **Gentle Follow-Up on Rebuttal for Submission 15942**
> >
> > Dear Reviewer rzYz,
> >
> > We hope this message finds you well. As the discussion period has been ongoing for several days, we would like to kindly follow up regarding our rebuttal to your initial review. We sincerely appreciate the time and effort you’ve dedicated to evaluating our work and would be grateful for your response for our rebuttal.
> >
> > Thank you again for your valuable feedback and for contributing to the discussion.
> >
> > Best wishes,\
> > Authors of Submission 15942

---

> > > ### Comment · Reviewer_rzYz · 2025-08-07
> > > **Score retained**
> > >
> > > Thank you for the detailed and thoughtful rebuttal.
> > > Regarding the first two points (prompt set and alpha sweet-spot), I appreciate the additional experiments you conducted to address these concerns.
> > > On the dimensionality of the FASA metric, thank you for the clarification, it was helpful in forming a more complete understanding.
> > > Concerning the FLUX variant of FASA, I believe it is important to acknowledge the existence of this alternative formulation, and ideally include a brief description in the main text, as you did in your response.
> > > After reading the other reviews and considering the rebuttal, I have decided to retain my original score.

---

> > > > ### Author Response · Authors · 2025-08-07
> > > > **Thank you!**
> > > >
> > > > Dear Reviewer rzYz,
> > > >
> > > > Thank you for your thoughtful and constructive feedback. We sincerely appreciate the time and effort you have dedicated to reviewing our work. In response to your valuable suggestions, we will make the following improvements in our updated submission:
> > > >
> > > > **Additional Experiments**. We will add our experimental analysis to include results from an extended prompt set and a fine-grained study of the parameter $\alpha$.
> > > >
> > > > **Enhanced Explanation of FASA**. To improve clarity, we will incorporate a more detailed discussion of FASA on DiTs.
> > > >
> > > > Once again, we are grateful for your insightful comments, which have significantly strengthened our paper.
> > > >
> > > > Best regards,\
> > > > Authors of Submission 15942

---

### Official Review · Reviewer_uVNV · 2025-07-08

**Clarity:** 2
**Significance:** 3
**Originality:** 3
**Rating:** 4
**Confidence:** 4

**Summary:**

This paper addresses the problem of human face personalization in text guided image generation. First, to understand the problem modes, two image generation pipelines are explored: personalized denoising, with the identity embedding incorporated into the denoising/generation process, and foundation denoising, where the identity embedding is set to zero. It is found that the PD model encodes identity but loses consistency with other components of the prompt. On the other hand, the FD model does not have identity but allows for consistency to the rest of the prompt. Further, observing the cross attention layers indicates that the identity embedding reduces the intensity of the cross attention maps for other text prompts.

To solve this, FreeCure is proposed. This training-free method consists of two parts: (a) Foundation-Aware Self Attention: a dual-inference pipeline is used, where attention maps are generated for both the FD and PD pathways. Next, an external masking model like segment anything generates segmentation masks for attributes corresponding to various different parts of the prompt. These masked attention maps from the FD branch are then concatenated and combined together in the final attention branch. (b) Asymmetric prompt guidance: after attributes with clear location restored are generated via FASA, APG enhances attributes through inversion and denoising. First, during inversion, only a template prompt (a man) is given. Then, during denoising, the full prompt with attributes is included. No identity embeddings are used in this process, only text.

Across both celebrity and non-celebrity identities, compared with several recent baseline methods, the proposed method is found to improve all prior methods in terms of identity and prompt consistency, across a range of metrics. The method is also found to work with several attributes.

**Questions:**

Questions:
1. I was not able to find the code in the submission: will that be included later?
2. How sensitive is the method to hyper parameters such as \omega and \gamma: are these the same for all methods?
3. In general, what types of models can the proposed method work with and what can it not work with? It would be good to include these in the limitations.

**Ethical Concerns:**

["NO or VERY MINOR ethics concerns only"]

**Final Justification:**

I would like the thank the authors for their response and the discussion. Based on this I would like to weight my score closer to acceptance, and recommend a weak accept score, taking into account both the strengths and the inherent weaknesses of this work.

**Limitations:**

Yes, with some suggestions above

**Quality:**

3

**Strengths And Weaknesses:**

Strengths:
1. The paper is largely well written, and method is easy to understand
2. The method is well motivated, with analyses pointing out the tradeoff between prompt consistency and identity
3. The method is simple, plug and play and easy to apply to most existing methods to improve prompt consistency
4. Results show benefits afforded to existing methods by including the proposed method

Weaknesses:
1. I could not find a discussion on runtime and computational overhead. Naively, it seems like there would be a conisderable overhead to each method due to the proposed method, that should be discussed.
2. It seems that the method would be quite sensitive to the accuracy of the mask generation method. How sensitive is the proposed method to inaccurate/partially accurate masks?
3. For qualitative heavy results such as this, it would be good to include a user study to emphasize the improvements afforded by the proposed method.

---

> ### Author Rebuttal · Authors · 2025-07-30
>
> We sincerely appreciate your valuable feedback and constructive suggestions, particularly the positive assessment of our paper's motivation, methodological simplicity, and comprehensive experimental analysis. In response to the raised concerns, we would like to address the following points for clarification.
> ## 1. Runtime Analysis
> We conduct experiments on a single H800 GPU with 80 GB VRAM to measure each baseline’s inference time before and after applying FreeCure. The table below reports the results ("*" means 30-step denoising due to official guidance and 50-step denoising is applied for the rest baselines):
> >**Table 1: Runtime Analysis**
> |     | Baseline (seconds) | Baseline + FreeCure (seconds) | |   |  |
> |:----:|:--------:|:---------:|:------:|:-------:|:--------:|
> |     |  **Total**| Stage 1| Stage 2 FASA | APG   | **Total**  |
> | FastComposer |**1.79**| 1.96| 2.19| 1.83  | **5.98** |
> | Face-Diffuser   |**2.13**| 2.63| 2.33 | 1.92  | **6.88**|
> | Face2Diffusion* |**1.13**| 1.25| 1.47| 1.23  | **3.95** |
> | PhotoMaker     |**6.52**| 7.04| 10.03| 6.1   | **23.17** |
> | InstantID*          |**7.12**| 7.63| 10.65| 6.23  |**24.51** |
> | PuLID-SDXL    |**7.43**| 8.02| 10.96| 7.28  | **26.26**  |
> | PuLID-FLUX*    |**12.84**| 14.89| 16.86| 12.95 | **44.7**  |
> | InfineteYou*      | **8.34**| 10.23| 13.53| 9.03  | **32.79**  |
>
> While we acknowledge that introducing extra calculation can lead to longer runtime, we also emphasize that most training-free methods introduce extra processes, including inversion operation, denoising processes, and extra attention calculation. Plus, comparing to the fact that most encoder-based personalization methods require a long range time of data collection, preprocessing and tuning (this may consume a large array of GPUs and thousands of GPU-hours), our training-free method provides an innovative perspective for prompt consistency improvement by the knowledge in themselves, without the need for designing new model architecture and dataset curation.
>
> ## 2. Robustness under the Scenario of Inaccurate Masks
> We reflect this concern by designing two-way validation experiments that introduce inaccuracy to attribute masks. First, we apply a **dilation** operation on masks, which might include area which does not belong to the target attribute area. Second, we apply an **erosion** operation, which might cause loss of some useful information about attributes. To create significant distortion of masks, we set the receptive field of both dilation and erosion to 25 pixels. We conduct experiments on several baselines, and the table below reports the results for respective prompt consistency (PC) and ideneity fidelity (IF):
> > **Table 2: PC and IF Performance for Inaccurate Masks**
> |            | PC with original masks | IF with original masks | PC with dilated masks | IF with dilated masks | PC with eroded masks | IF with eroded masks |
> |---|----|------|---|----|----|----|
> | InstantID  | 23.62    | 62.01   | 23.45  | 62.23  | 23.21  | 62.23  |
> | PhotoMaker | 24.91   | 50.15   | 24.87  | 50.01    | 24.34 | 50.44   |
> | PuLID-SDXL | 26.05  | 56.95  | 25.51  | 56.34 | 25.36  | 57.01 |
> | PuLID-FLUX | 24.78   | 72.61  | 24.52   | 72.39   | 24.32  | 72.97  |
>
> The results demonstrate that identity fidelity (IF) remains largely consistent across both dilated and eroded mask conditions, with slight improvements observed in eroded masks. This occurs because mask erosion preserves certain attributes similar to the reference image. As for prompt consistency, both dilation and erosion cause minimal degradation, though erosion exhibits marginally greater impact. We consider that this stems from cases where fine-grained attribute restoration (e.g., iris color, earrings) may fail when the mask completely degenerates due to erosion.
>
> Generally, the performance degradation of FreeCure due to inaccurate masks is limited. Moreover, in practical applications, specialized segmentation models (e.g., BiSeNet, Segment Anything) exhibit high robustness, making failure cases caused by mask inaccuracies even rarer than observed in these deliberately designed experiments. This further validates the inherent robustness of the FreeCure framework.
>
> ## 3. User Study
> We appreciate your attention to this point and agree that a user study can provide constructive support for the evaluation of our method. We conduct an online questionnaire with 30 participants. Regarding the questionnaire design: For each baseline, we randomly select 10 samples. Each sample consists of a reference image, a prompt, the baseline-generated result, and the result after applying FreeCure. Participants are asked to evaluate the identity fidelity (IF) and prompt consistency (PC) of each result. Specifically, they rate their satisfaction on a scale of 1 to 10, considering identity fidelity and prompt consistency. The survey samples from different baselines are presented randomly, and the order of the baseline results and FreeCure-applied results is also randomized to ensure that participants' judgments for each sample remained relatively independent. Based on scoring results, we provide both the scoring average values and preference ratio.
> > **Table 3: User Study Score**
> |     | Baseline      |    | Baseline + FreeCure |    |
> |--|----|-----|-----|---|
> |     | IF Score Avg. | PC Score Avg. | IF Score Avg.       | PC Score Avg. |
> | FastComposer   | 7.55  | 4.96   | 7.51 | **8.35** |
> | Face-Diffuser     | 7.56  | 5.08   | 7.41  | **7.62**   |
> | Face2Diffusion   | 7.83  | 5.02   | 7.71  | **7.83**  |
> | PhotoMaker       | 6.95  | 6.41   | 7.49   | **8.04**   |
> | InstantID            | 7.78  | 5.28   | 7.79   | **8.20**   |
> | PuLID-SDXL      | 7.76  | 6.38   | 8.08    | **8.26**   |
> | PuLID-FLUX      | 7.78  | 6.56   | 8.08   | **8.02**  |
> | InfineteYou        | 7.83  | 6.52   | 7.79   | **8.18**  |
>
> > **Table 4: User Preference on Identity Fidelity**
> |   | Prefer Baseline | Prefer Equally | Prefer Baseline + FreeCure |
> |---|----|---|-----|
> | FastComposer   | 28.88% | **60.54%**  | 10.58%  |
> | Face-Diffuser     | 25.33%  | **63.83%** | 10.83%|
> | Face2Diffusion   | 18.88%  | **68.00%** | 13.13%  |
> | PhotoMaker       | 12.54% | **78.83%** | 8.63%  |
> | InstantID            | 25.38% | **64.29%**  | 10.33%  |
> | PuLID-SDXL      | 10.67% | **81.38%**  | 7.96%  |
> | PuLID-FLUX      | 11.50% | **84.67%** | 3.83%  |
> | InfineteYou        | 10.67% | **79.25%**   | 10.08%   |
>
> > **Table 5: User Preference on Prompt Consistency**
> |                | Prefer Baseline | Prefer Equally | Prefer Baseline + FreeCure |
> |---|--|---|---|
> | FastComposer   | 4.29%  | 14.38% | **81.33%** |
> | Face-Diffuser  | 5.17%  | 16.71%   | **78.13%**  |
> | Face2Diffusion | 4.04%   | 16.04%| **79.92%**    |
> | PhotoMaker     | 4.38%   | 31.88%  | **63.75%**  |
> | InstantID      | 8.46%    | 12.58%  | **78.96%**  |
> | PuLID-SDXL     | 4.29%  | 16.92% | **78.79%** |
> | PuLID-FLUX     | 10.54%  | 16.46%  | **73.00%**  |
> | InfineteYou    | 9.75%   | 21.00% | **69.25%** |
>
> The results show that, based on human evaluation, FreeCure consistently improves every baseline's prompt consistency. Furthermore, we observe that there are many cases where participant tends to give the same score for identity fidelity. Such performance supports the fact that FreeCure's negative effect on identity fidelity is quite limited in a practical manner and promises its practical use in the future. In our updated version, we plan to provide the template example for the questionnaire and more straightforward visualization techniques (e.g., column chart) to demonstrate the results of the user study.
>
> ## 4. Hyper-parameter analysis
> We acknowledge that the optimal configuration of FreeCure varies slightly for each baseline. However, this does not constitute a weakness: for each baseline, the fluctuation of their optimal hyperparameters is within an acceptable range:
> $$
> \omega \in [1.8, 2.4], \gamma \in [0.5, 0.6]
> $$
> Baselines built on Stable Diffusion v1.5 (Face2Diffusion, FastComposer, Face-Diffuser) require a larger $\omega$ value up to 2.4, and other baselines remain around 2.0. Within this range, the overall results do not differ much, which allows us to determine a unified hyperparameter setting without any further complicated strategy ($\omega = 2.0$ and $\gamma = 0.5$ throughout all experiments in the submission), and still achieved consistent performance improvement in prompt consistency with the maintenance of identity fidelity.
>
> Secondly, we would like to point out that since the foundation models and architectures of each baseline are diverse, FreeCure—being a plug-and-play method—should be allowed for a reasonable adjustable range to better adapt to each baseline.
>
> ## 5. Code Release
> Thanks for pointing this out. We will release the code of FreeCure later.
>
> ## 6. Limitations
> Our method can seamlessly apply to most diffusion-based personalization models (including those based on SD-v1.5, SDXL, and DiT), which dominate the personalization task. Since currently there is no influential personalization method based on auto-regressive models, it may cost some extra effort to adapt our methods to these methods in the future. We will add the discussion of this matter in the limitations.

---

> > ### Comment · Reviewer_uVNV · 2025-08-03
> > **Thank you for the response**
> >
> > Dear authors,
> >
> > Thank you very much for your response to my questions. Largely, my questions have been addressed. Just want to note the following points:
> >
> > 1. Thank you for the runtime analysis. It seems the proposed method increases the runtime by, on average, 3.5x. This is a significant overhead. While I recognize the argument made by the authors that this is a tradeoff vs training-based methods that need significant compute at training time, as well as that the tradeoff of increased runtime overhead vs better prompt fidelity is valuable for follow-up work, I still think this significant increased runtime at inferences is an aspect of the method that I hope the authors discuss in the revision.
> > 2. I appreciate the user study, it would be great if the authors include this in the revision to further strengthen their claims.
> > 3. In interest of reproducibility, it would be good if the authors can mention all the hyper parameters, for all the experiments, either in the main paper or the supplement.
> > 4. The authors mention that the code will be released ‘later’. I just want to confirm that this means the code will be released on acceptance of the paper.

---

> > > ### Author Response · Authors · 2025-08-04
> > >
> > > Dear Reviewer uVNV,
> > >
> > > Thanks for your thoughtful feedback and for acknowledging our responses to your initial questions. We sincerely appreciate your time and constructive suggestions. Below, we address your additional points in detail:
> > >
> > > + **Runtime**. In the updated paper, we will expand the discussion in the Limitations section to analyze this trade-off. Based on this, we'll provides some thoughts about future work for optimization.
> > >
> > > + **User Study**. We appreciate your suggestion and will incorporate the user study results into the main submission (or supplementary materials) to further validate our claims. The findings will be presented alongside quantitative metrics to provide a comprehensive evaluation. The template of user study will also be provided, as space permits.
> > >
> > > + **Hyperparameters**. We will ensure all hyperparameters for experiments are clearly documented, either in the main paper (if feasible) or in the supplementary material, to facilitate reproducibility.
> > >
> > > + **Code Release**. Yes, we confirm that the code will be made publicly available upon acceptance of the paper. We will also include a note in the paper to clarify this timeline.
> > >
> > > Again, thank you once again for your invaluable time and comments that have significantly strengthened our work.
> > >
> > > Best wishes,\
> > > Authors of Submission 15942

---

> > > ### Author Response · Authors · 2025-08-05
> > > **Gentle Follow-up before Final Rating**
> > >
> > > Dear Reviewer uVNV,
> > >
> > > We hope this message finds you well. We are pleased to hear that our rebuttal has largely addressed your concerns, and we truly appreciate your constructive feedback. As mentioned previously, we highly value your suggestions and will incorporate them into our updated submission.\
> > > Before the final rating is determined, we want to ensure we remain open to any further thoughts or concerns you may have regarding our work. Please don’t hesitate to reach out if there’s anything additional you’d like to discuss. Once again, thank you for your time and effort throughout the review and discussion process that have greatly improved our paper.
> > >
> > > Best regards,\
> > > Authors of Submission 15942

---

### Author Response · Authors · 2025-08-09
**Summary of Rebuttal and Discussion**

Dear Reviewers and Area Chairs,

We sincerely thank all reviewers for their valuable time, constructive comments, and active engagement during the discussion period. We are greatly encouraged that all five reviewers have expressed a willingness to maintain or revise their ratings positively, and we deeply appreciate the constructive spirit of the entire exchange.

Specifically:

+ **Ethics Reviewer ZvZ3** confirmed that all issues regarding data privacy and ethical risks have been addressed.

+ **Reviewer uVNV**'s concerns have been addressed, and the reviewer appreciated the supplemental user study experiment, which effectively supported our proposed method.

+ **Reviewer rzYz** acknowledged the experiments about empirical study and additional explanation of some technical details and retained a positive rating of "Accept".

+ **Reviewer nf3e** appreciated our efforts during the rebuttal and raised the rating to 'Borderline Accept.'

+ **Reviewer kayu** confirmed that all concerns have been addressed and recommended revising the rating to positive.

+ **Reviewer tTVJ** provided a positive initial rating in the first review, and the reviewer’s concerns largely overlapped with those of other reviewers—all of which were effectively addressed.

We have provided detailed responses to each reviewer and summarized the key supplementary experiments and additional clarifications below.

## Additional Experiments

+ **User Study**: We conducted an extensive user study through an online questionnaire. The results demonstrate that FreeCure significantly improves prompt consistency across all personalization models while maintaining minimal impact on identity consistency.

+ **Robustness on Inaccurate Mask**: We apply erosion and dilation operations to the extracted masks, which have minimal impact on our method, demonstrating FreeCure's robustness.

+ **Diverse Prompt**: We extended the prompt attributes beyond facial features (e.g., clothing, headphones, cups), demonstrating FreeCure's superior generalization capability.

+ **Experiment about Empirical Study**: We present a fine-grained analysis of $\alpha$ (introduced in **Section 3**), demonstrating the limitations of cross-attention manipulation and further motivating the need for FASA.

+ **Runtime**: We reported the runtime of FreeCure.

## Clarifications and Explanations

+ **Implementation on FLUX**: We provided detailed information about FASA's implememtation on FLUX, with clear formula.

+ **Hyper-parameter**: We confirmed that all experiments in the submission were conducted using the same settings for  $\omega$ and $\gamma$, demonstrating FreeCure's generalizability.

+ **Automation of Mask Extraction**: We provided details of how semgentation models automatically extract masks of different attributes.

+ **Difference between Related Methods**: We provided the core difference of FreeCure between CFG, MasaCtrl and MoA.

We hope these points highlight the rigor and impact of our work. Thank you again for your time and thoughtful consideration.

Best wishes,\
Authors of Submission 15942

---

### Decision · Program_Chairs · 2025-09-17

**Decision:**

Accept (poster)

**Comment:**

This paper introduces FreeCure, an innovative and practical training-free framework designed to tackle a well-known trade-off in facial personalization: maintaining a subject's identity while accurately following textual prompts. The authors point out that strong identity embeddings can suppress the model's ability to render other attributes. Their solution leverages a dual-inference paradigm, using the foundation model's superior attribute control to guide the personalized generation process without any retraining. The method is generally simple, effective, and broadly applicable to popular models like Stable Diffusion and FLUX. Given the strong empirical analysis, positive results, applicability to broad models, and the overall positive feedback from the reviews after a thorough discussion period, this paper gets a clear accept recommendation from me.

I would also like to note that the paper was flagged for an ethics review regarding the source of its non-celebrity dataset and a discussion of potential misuse, but the authors successfully addressed these points by clarifying their data sources and expanding the limitations section, satisfying the ethics reviewer. I'm confident that the final camera-ready version, incorporating the thoughtful suggestions from all reviewers, will be a strong and well-rounded contribution to the conference.